# Assessing the mechanism and therapeutic potential of modulators of the human Mediator complex-associated protein kinases

Paul A Clarke[1]*[†], Maria-Jesus Ortiz-Ruiz[1][†], Robert TePoele[1], Olajumoke Adeniji-Popoola[1], Gary Box[1], Will Court[1], Stephanie Czasch[2], Samer El Bawab[2], Christina Esdar[2], Ken Ewan[3], Sharon Gowan[1], Alexis De Haven Brandon[1], Phillip Hewitt[2], Stephen M Hobbs[1], Wolfgang Kaufmann[2], Aurélie Mallinger[1], Florence Raynaud[1], Toby Roe[1], Felix Rohdich[2], Kai Schiemann[2], Stephanie Simon[2], Richard Schneider[2], Melanie Valenti[1], Stefan Weigt[2], Julian Blagg[1], Andree Blaukat[2], Trevor C Dale[3]*, Suzanne A Eccles[1], Stefan Hecht[2], Klaus Urbahns[2], Paul Workman[1]*, Dirk Wienke[2]*

[1]Cancer Research UK Cancer Therapeutics Unit, The Institute of Cancer Research, London, United Kingdom; [2]Merck KGaA, Darmstadt, Germany; [3]School of Bioscience, Cardiff University, Cardiff, United Kingdom

*For correspondence: paul. clarke@icr.ac.uk (PAC); daletc@ cardiff.ac.uk (TCD); paul. workman@icr.ac.uk (PW); dirk. wienke@merckgroup.com (DW)

[†]These authors contributed equally to this work

**Abstract** Mediator-associated kinases CDK8/19 are context-dependent drivers or suppressors of tumorigenesis. Their inhibition is predicted to have pleiotropic effects, but it is unclear whether this will impact on the clinical utility of CDK8/19 inhibitors. We discovered two series of potent chemical probes with high selectivity for CDK8/19. Despite pharmacodynamic evidence for robust on-target activity, the compounds exhibited modest, though significant, efficacy against human tumor lines and patient-derived xenografts. Altered gene expression was consistent with CDK8/19 inhibition, including profiles associated with super-enhancers, immune and inflammatory responses and stem cell function. In a mouse model expressing oncogenic beta-catenin, treatment shifted cells within hyperplastic intestinal crypts from a stem cell to a transit amplifying phenotype. In two species, neither probe was tolerated at therapeutically-relevant exposures. The complex nature of the toxicity observed with two structurally-differentiated chemical series is consistent with on-target effects posing significant challenges to the clinical development of CDK8/19 inhibitors.

## Introduction

The Mediator complex is a multi-subunit regulator of transcription in eukaryotes that transfers signals from DNA-bound transcription factors to the RNA polymerase II pre-initiation complex (*Allen and Taatjes, 2015*; *Yin and Wang, 2014*; *Poss et al., 2013*). It also has a role in transcription elongation and pausing, and can influence chromatin structure, where it facilitates the formation of enhancer-promoter gene loops and is enriched at 'super-enhancer' regions (*Allen and Taatjes, 2015*; *Poss et al., 2013*; *Whyte, 2013*). Mediator activity is regulated by the association with a four-subunit kinase module containing cyclin-dependent kinase 8 (CDK8), cyclin C (CCNC) and Mediator subunits MED12 and MED13 (*Allen and Taatjes, 2015*; *Poss et al., 2013*; *Taatjes et al., 2002*). As a kinase that reversibly associates with the Mediator, CDK8 is thought to regulate gene expression through phosphorylation of transcription factors and Mediator subunits (*Rzymski et al., 2015*).

**eLife digest** Healthy cells in the human body can become cancerous if they gain genetic mutations that allow them to rapidly grow and divide. Some types of cancer respond better to drug treatments than others and tumors often develop resistance to a particular drug treatment after a while. Because of this, researchers are always searching for new molecules to develop into anticancer drugs.

Recently, a team of researchers identified some small molecules that could inactivate two closely related proteins called CDK8 and CDK19. CDK8 is essential for the WNT signaling pathway – which enables cells to communicate with one another – and has been extensively studied in various cancers. Previous studies indicate that this protein can either promote or inhibit the growth of tumors, depending on the type and stage of the cancer. Furthermore, CDK8 regulates a type of molecular switch called a "super-enhancer", which controls the activity of many genes. In contrast, the role of CDK19 in cells was not as well understood. Here Clarke, Ortiz-Ruiz et al. investigated whether two different classes of small molecules that target CDK8 and CDK19 (referred to as "prototype CDK8/19 drugs") could inhibit the growth of cancers, and whether they have any harmful side effects on healthy cells.

For the experiments, human cancer cells were implanted into mice. Treating these mice with prototype CDK8/19 drugs inhibited the activity of CDK8 and CDK19 in the cancer cells and slowed the growth of colorectal tumors. A type of blood cancer called acute myeloid leukaemia was particularly sensitive to the drugs. However, Clarke, Ortiz-Ruiz et al. also observed that the prototype drugs altered the activity of many genes with roles in healthy tissues such as immune, bone and stem cells. Further experiments in mice and cells grown in the laboratory confirmed that these prototype drugs have adverse effects on healthy intestinal and bone marrow stem cells and trigger changes to immune cells. These concerning side effects were also evident when the prototype drugs were tested in rats and dogs. Furthermore, the experiments indicate that there is not a suitable range of doses of these drugs in which the therapeutic benefits outweigh the toxic side effects.

Clarke, Ortiz-Ruiz et al. conclude that the clinical development of CDK8/19 drugs will be extremely challenging and that their prototype drugs would not currently be suitable for use as cancer treatments. However, the small molecules they describe will be important probes in research to study exactly how CDK8/19 regulate gene activity in both healthy cells and cancers.

Phosphorylation by CDK8 can directly alter transcription factor activity (*Bancerek et al., 2013*; *Morris et al., 2008*; *Zhao et al., 2013*) or mark factors for degradation (*Fryer et al., 2004*; *Alarcón et al., 2009*; *Zhao et al., 2012*; *Li et al., 2014*). The role of CDK8, and the Mediator kinase module, in the control of transcription may not be unique as paralogs of CDK8, MED12 and MED13 have been identified that may have distinct roles in vitro and in vivo (*Sato et al., 2004*; *Tsutsui et al., 2008*; *Galbraith et al., 2013*; *Westerling et al., 2007*).

The biological function of CDK8 varies by cell type and response to different stimuli (*Allen and Taatjes, 2015*; *McCleland et al., 2015*). This is particularly true in cancer, where *CDK8* may function not only as an oncogene, but also as a tumor-suppressor depending on the cellular context (*McCleland et al., 2015*; *Mitra et al., 2006*; *Chattopadhyay et al., 2010*; *Gu et al., 2013*; *Firestein et al., 2008, 2010*; *Seo et al., 2010*; *Adler et al., 2012*). *CDK8* may act as an oncogene in colorectal cancer where *CDK8* is amplified, with copy number gains observed in ~60% of tumors (*Firestein et al., 2010*; *Seo et al., 2010*), and shRNA knockdown can reduce the growth of human colorectal cancer xenografts harbouring *CDK8* gene amplification (*Firestein et al., 2008*; *Adler et al., 2012*; *Starr et al., 2009*). Furthermore, *CDK8* expression is reportedly required for growth of colorectal cancer xenografts and to maintain embryonic stem cells in an undifferentiated state (*Adler et al., 2012*). Importantly, *CDK8* expression transforms fibroblasts into a malignant phenotype, whereas expression of a kinase-dead mutant does not (*Firestein et al., 2008*). An shRNA screen has also demonstrated a requirement for CDK8 in the activation of WNT signaling in

colorectal cancer (*Firestein et al., 2008*), suggesting that CDK8 and the Mediator kinase module may promote oncogenesis through activation of the canonical WNT pathway.

Previously, we reported the discovery and optimization of a potent and selective 3,4,5-trisubstituted pyridine series of small-molecule inhibitors of WNT signaling from a cell-based pathway screen, and using a chemo-proteomic strategy we identified CDK8 and CDK19 as the primary molecular targets (*Dale et al., 2015*; *Boyer, 2015*). Through further optimization we identified a potent, highly selective and orally bioavailable dual CDK8/19 ligand with excellent cell-based activity and pharmaceutical properties (*Mallinger et al., 2016a*). Subsequently, we discovered a second, chemically-distinct series of CDK8/19 ligands and optimization of pharmacological, pharmaceutical and pharmacokinetic properties identified a 3-methyl-1*H*-pyrazolo[3,4-*b*]pyridine, which also binds to CDK8/CCNC (*Czodrowski et al., 2016*). With potent and selective exemplar compounds from these two structurally differentiated chemical series in hand together with corresponding inactive control compounds, we were well positioned to investigate the therapeutic potential of dual CDK8/19 modulation. Specifically, we set out to establish if these compounds had antiproliferative or antitumor activity and whether a therapeutic window could be identified in preclinical models that would justify the clinical development of these compounds.

## Results

### Characterisation of structurally differentiated CDK8/19 ligands

We identified two structurally differentiated, potent, selective and cell permeable chemical series, namely 3,4,5-trisubstituted pyridines and 3-methyl-1*H*-pyrazolo[3,4-*b*]pyridines, suitable for exploring the function of the Mediator complex-associated protein kinases CDK8 and CDK19 (*Figure 1A* and *Figure 1—source data 1*). In addition to two tool compounds, 1 (CCT251545; *Mallinger et al., 2015*) and 2 (compound 42; *Mallinger et al., 2016a*), that fulfill all of the criteria set out for chemical probes (*Frye, 2010*), the lead compounds from each of the chemical series, 3 (CCT251921; *Mallinger et al., 2016a*) and 4 (MSC2530818; *Czodrowski et al., 2016*) had optimal pharmacological and pharmaceutical properties that made them suitable for further progression to preclinical studies (*Figure 1A* and *Figure 1—source data 1*). All four compounds had single digit nanomolar binding affinities for CDK8 and 19, and were very highly selective with little evidence for off-target activity in extended protein kinase panels (*Figure 1—source data 1*). Our compounds also potently inhibited inducible (7dF3; *Ewan et al., 2010*) or basal (LS174T; *Dale et al., 2015*; *Mallinger et al., 2015*) WNT-pathway luciferase-reporter expression together with STAT1$^{SER727}$ phosphorylation  a target-engagement biomarker  at low nanomolar concentrations (*Figure 1A* and *Figure 1—source data 1*) (*Bancerek et al., 2013*; *Dale et al., 2015*).

A comparison of the co-crystal structure of CDK8/CCNC with 3 or 4 showed that both molecules adopt a Type I binding mode and make similar contacts with active site residues (*Figure 1B*). Compound 3 binds in a twisted conformation, as previously described for 1, with the indazole substituent at C5 of the pyridine ring forming a pi-cation interaction with Arg356 (*Figure 1B*) (*Mallinger et al., 2015*, *2016a*). Compound 4 forms similar interactions with the hinge region and with the catalytic Lys52 to those observed for compound 3, and its *p*-chlorophenyl substituent occupies the same region as the indazole subsituent of compound 3; however, the scaffold architecture of the two compounds is entirely different (*Figure 1B*). Throughout our studies on both chemical series, we observed a strong correlation between the compounds' affinities for both CDK8 and CDK19, suggesting that selective inhibition of CDK8 versus CDK19 is likely to be a significant challenge (*Dale et al., 2015*; *Mallinger et al., 2016a*). This reflects the high sequence similarity between CDK8 and CDK19 (*Figure 1C*). We also tested selected compounds from a further three chemical series that we identified from the literature and again could not detect any substantial selectivity for CDK8 versus CDK19 (*Figure 1—source data 1*).

### In vitro activity of chemical probes and preclinical candidates

We confirmed compound activity in a range of in vitro assays using human colorectal cancer cell lines, including some with an increased *CDK8* gene copy number (*Figure 1—source data 1*). All four compounds potently inhibited a WNT-dependent reporter in all of the cell lines tested, but did not inhibit a WNT-independent housekeeping *EEF1A1* promoter-reporter construct in the negative

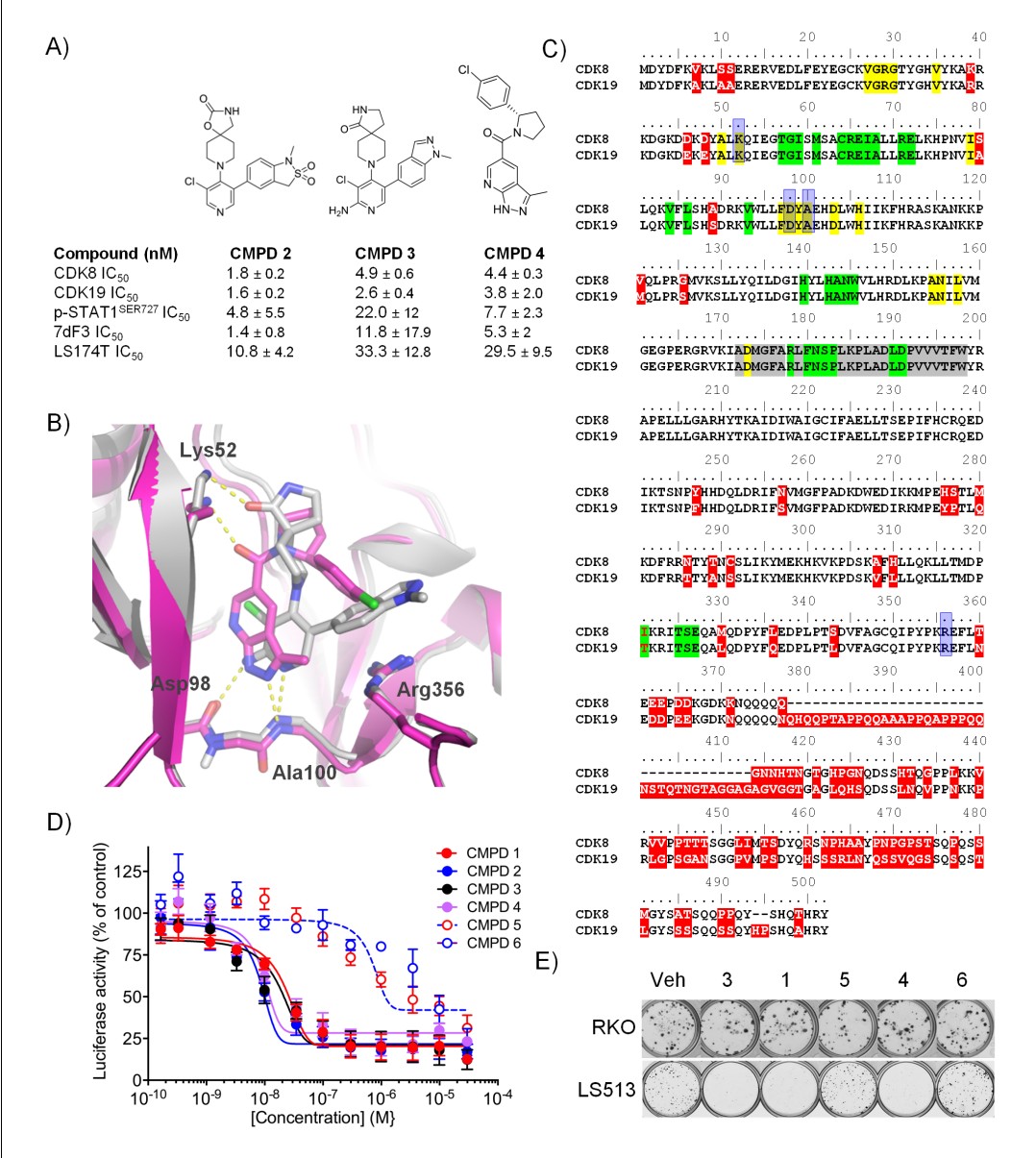

**Figure 1.** Optimised compounds for exploring CDK8 and CDK19 function. (A) Chemical structure and activity of compounds 2, 3 and 4 (n > 2, mean ± s.d.). (B) Overlay of 3 (grey; ocd 5HBJ) and 4 (pink; Pdb code: 5IDN) bound to CDK8/CCNC. Key interactions (yellow) and residues are shown. Residues 23 – 39 and 359 – 361 have been cropped for clarity. (C) Amino acid sequence alignment for human CDK8 and CDK19. Red, sequence differences; yellow, ATP binding; green, CCNC binding; gray, activation loop; blue, inhibitor binding. (D) Luciferase activity in COLO205-cl4 cells containing a TCF/LEF reporter gene construct following 24 hr compound treatment (n = 4, mean ± s.d.). (E) Colony assay. Plates were seeded with LS513 or insensitive RKO cells and treated for 14 d.

The following source data and figure supplements are available for figure 1:

**Source data 1.** Properties of CDK8/19 ligands and their effects on reporter expression and cell proliferation in a human colorectal cancer cell line panel.
**Figure supplement 1.** Effect of CDK8 and CDK19 shRNA and siRNA treatment in CDK8-amplified human colorectal cancer cell lines.
**Figure supplement 2.** Comparison of CDK8 and CDK19 gene copy number or protein expression with sensitivity to treatment with compound.

control RKO colorectal cancer cell line, which expresses low levels of beta-catenin (*Figure 1D* and *Figure 1—source data 1*) (*Mallinger et al., 2015*; *Dale et al., 2015*). Weakly-active negative-control compounds from the 3,4,5-trisubstituted pyridine and 3-methyl-1*H*-pyrazolo[3,4-*b*]pyridine series (compounds 5 and 6 respectively) did not inhibit reporter gene expression or STAT1$^{SER727}$ phosphorylation (*Figure 1—source data 1*). Given the potent inhibition of reporter activity we were surprised by the lack of effect of our potent inhibitors on tumor cell proliferation after standard 4 d continuous exposure conditions (*Figure 1—source data 1*). However, this was consistent with a reported lack of antiproliferative effects for a different chemical series in a single colorectal cancer cell line (*Koehler et al., 2016*). Silencing of *CDK8* and/or *CDK19* by shRNA in *CDK8*-amplified COLO205 cells also had no effect on viability, despite evidence for inhibition of reporter output or target gene expression (*Figure 1—figure supplement 1A*). Knockdown was more effective when we used *CDK8* and/or *CDK19* siRNA in *CDK8*-amplified HT29 cells, but again we saw no significant effect on viability after 5 d exposure to siRNA, despite near complete inhibition of STAT1$^{SER727}$ phosphorylation by the *CDK8* siRNA (*Figure 1—figure supplement 1B–C*). In contrast, a 14 d colony growth assay revealed a significantly similar antiproliferative effect for the lead compounds from both chemical series (p<0.001 for all comparisons with 1, 3 and 4; *Figure 1—figure supplement 2*), which was not observed for the negative-control compounds 5 and 6. However, no compounds showed colony growth inhibition in the negative control RKO colorectal cancer cell line (*Mallinger et al., 2015*; *Dale et al., 2015*). In this assay, we found three beta-catenin mutant (LS513, LS180, LS174T) and an APC mutant (SW620) colorectal cell lines to be most sensitive to treatment (*Figure 1E* and *Figure 1—source data 1*). The association between beta-catenin mutation and sensitivity to compound treatment in the colony assay did not reach significance (p>0.05). The lack of response of the RKO cells suggested that colony growth in this line did not require beta-catenin or CDK8 and contrasted with the APC or beta-catenin mutant lines where colony growth appeared to be dependent on beta-catenin-regulated transcription that also required CDK8/19. Overall, there was no significant correlation between compound activity in TCF-reporter, phospho-biomarker, colony growth assays and either CDK8/19 protein levels or gene copy number (*Figure 1—figure supplement 2* and *Figure 1—source data 1*).

## CDK8/19 ligands have modest activity against human colorectal cancer tumor xenogafts

Next, we determined if our two series of compounds had antitumor activity in vivo in human colorectal cancer xenograft mouse models. Previously, we demonstrated that compound 1 inhibited TCF/LEF-reporter gene expression and reduced STAT1$^{SER727}$ phosphorylation by >80%. This translated into tumor growth inhibition following oral dosing of 1 in mice bearing established COLO205 or SW620 colorectal cancer cell xenografts (*Mallinger et al., 2015*; *Dale et al., 2015*). We also found evidence for a significant, dose-dependent, reduction in tumor growth in HCT116 human colorectal cancer cell line xenografts as well as significant tumor growth inhibition (TGI = 81%; p<0.001) at 70 mg/kg in an LS513 human colorectal cancer xenograft model with concomitant reduction of p-STAT1$^{SER727}$ at 6 hr (*Figure 2—figure supplement 1* and *Figure 2—source data 1*).

For the lead compounds 3 and 4, we modelled the inhibition of STAT1$^{SER727}$ phosphorylation using multiple sets of experimentally-derived data from HCT116 and SW620 human tumor xenografts (*Figure 2* and *Figure 2—figure supplement 2A–C*). Detailed analysis, initially in HCT116 tumor xenografts, showed that maximal inhibition of STAT1$^{SER727}$ phosphorylation required treatment with ≥5 mg/kg compound 3 and that higher concentrations prolonged the period of maximal inhibition (*Figure 2—figure supplement 2A–C*). In SW620 tumor xenografts biomarker inhibition was rapidly achieved following treatment with 3 or 4 and could be maintained for approximately 10 hr after a single treatment with 30 mg/kg of 3, while 30 or 100 mg/kg of 4 prolonged the period of inhibition of STAT1$^{SER727}$ phosphorylation so that, unlike 3, multiple dosing with 4 prevented recovery of biomarker to control levels between treatments (*Figures 2A,3D–E*, *Figure 2—figure supplement 2D* and *Figure 2—figure supplement 3B*). Both lead compounds 3 and 4 exhibited reproducible, dose-dependent antitumor activity in SW620 tumor xenografts (*Figure 2D–E*, *Figure 2—figure supplement 3A–B* and *Figure 2—source data 1*) and also in an additional LS1034 colorectal tumor model that responded to our compounds in the in vitro clonogenic assay (*Figure 2—figure supplement 3C–E*, *Figure 2—figure supplement 4* and *Figure 2—source data 1*). While 5 mg/kg compound 3 induced maximal inhibition of STAT1$^{SER727}$ phosphorylation in SW620

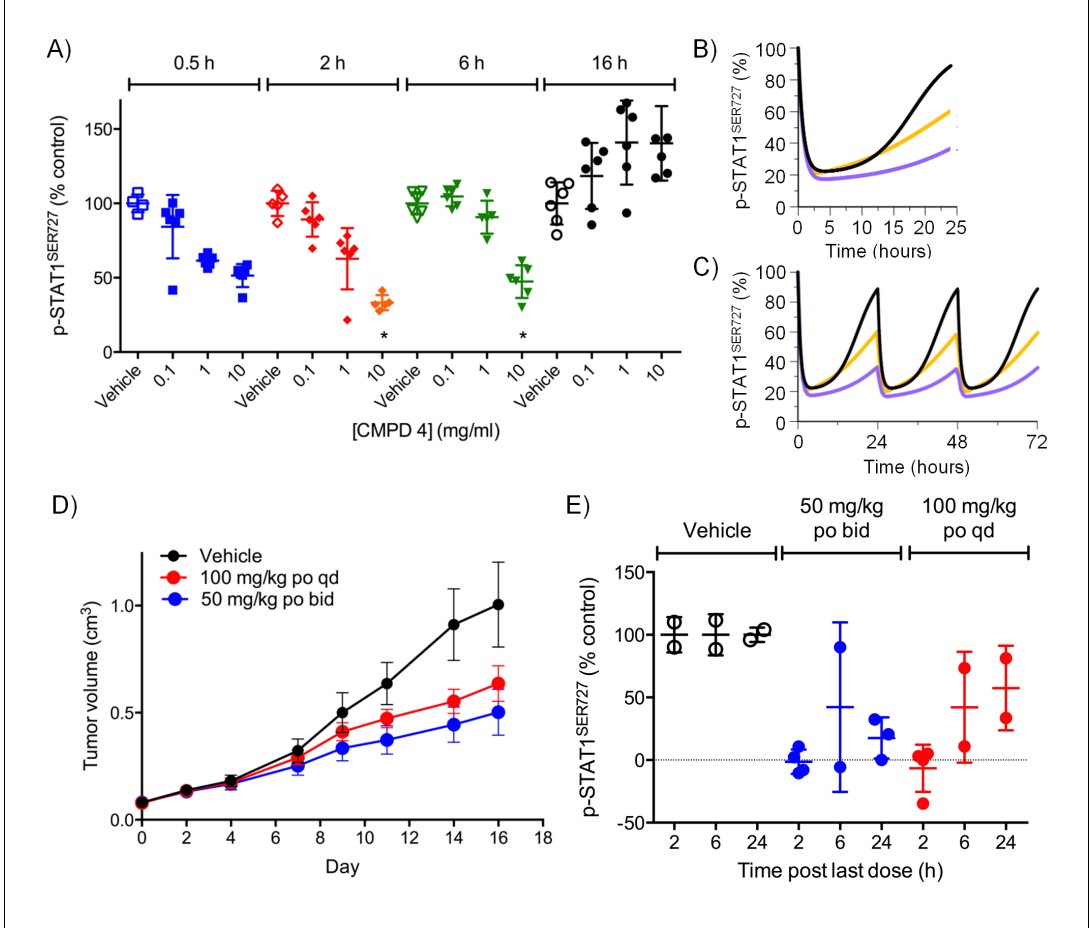

**Figure 2.** Target inhibition and antitumor activity of CDK8/19 ligands 3 and 4 in established human colorectal cancer cell line xenografts. (A) Level of p-STAT1$^{SER727}$ in SW620 human colorectal cancer xenografts following a single dose of 4, relative to the p-STAT1$^{SER727}$ level in vehicle-treated mice. Significance was determined by Kruskal-Wallis test and Dunn's post-test (*p=<0.001; ). (B, C) Modelling of experimental data, including data from (A) and *Figure 2—figure supplement 2D*, of STAT1$^{SER727}$ phosphorylation following (B) single or (C) twice daily doses of 30 mg/kg 3 (black) or 30 and 100 mg/kg 4 (orange and purple). (D) Volume of SW620 xenografts in mice treated with 4 or a vehicle control. (E) Level of p-STAT1$^{SER727}$, relative to control, in SW620 tumor xenografts at the stated time following the final dose of 4 (from D).

The following source data and figure supplements are available for figure 2:

**Source data 1.** Details of human colorectal cancer cell line xenograft studies.

**Figure supplement 1.** Differential antitumor activity in human LS513 colorectal cancer xenografts treated with CDK8/19 ligand 1.

**Figure supplement 2.** Pharmacodynamic profiling of CDK8/19 ligand 3 in human colorectal cancer xenografts.

**Figure supplement 3.** Antitumor activity and target engagement in human colorectal cancer xenografts treated with CDK8/19 ligands 3 and 4.

**Figure supplement 4.** LS1034 colony assay.

human tumor xenografts, its effects were short-lived, and were not sufficient to translate into antitumor activity (*Figure 2—figure supplement 2D* and *Figure 2—source data 1*). However, higher 30 mg/kg doses of compound 3 prolonged the maximal inhibition of STAT1$^{SER727}$ phosphorylation and had a significant (p<0.01) antitumor effect (*Figure 2—figure supplement 3A–B* and *Figure 2—source data 1*). As with 3, 10 mg/kg of 4 inhibited the pathway biomarker for 6 hr, but was not sufficient for antitumor activity in SW620 xenografts (*Figure 2A* and *Figure 2—source data 1*). However,

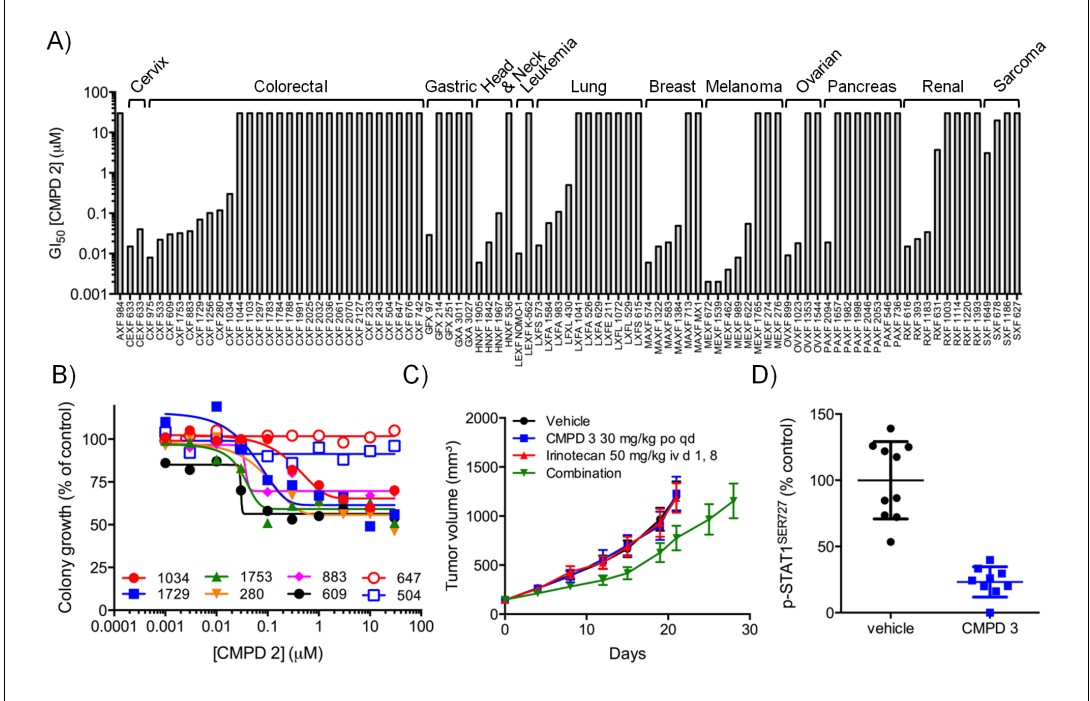

**Figure 3.** In vitro and in vivo activity of CDK8/19 ligands in patient-derived tumour xenograft models. (**A**) $GI_{50}$ values for 2 in PDX soft agar colony cultures. (**B**) Exemplar dose-response profiles for selected colorectal cancer clonogenic assays treated with 2. (**C**) Volume of human colorectal cancer CXF 1034 ($CTNNB1^{MUT}$, $PIK3CA^{MUT}$, $PTEN^{MUT}$) PDXs in mice treated with vehicle, 3 and / or irinotecan (mean values ± s.e.m., n = 10 per cohort). Tumor volume was significantly different (p=<0.001; 2 way ANOVA and Tukey's multiple comparison test) in mice receiving the combination treatment, compared with the monotherapy groups. (**D**) Level of p-STAT1$^{SER727}$, relative to control, in CXF 1034 xenografts in mice treated with 3 measured 1 hr after the final dose (p=<0.0001, Mann-Whitney test; mean values ± s.d., n = 10 per cohort).

The following figure supplements are available for figure 3:

**Figure supplement 1.** In vivo activity of CDK8/19 ligand 3 in PDXs.

**Figure supplement 2.** Pharmacodynamic and antitumor activity of 3 and 4 in AML models.

repeated dosing of 4 at 50 or 100 mg/kg prolonged the inhibition of STAT1$^{SER727}$ phosphorylation and gave evidence of antitumor activity (**Figure 2D–E** and **Figure 2—source data 1**). Hence, the antitumor activity of 3 and 4 against colorectal cancer xenografts showed clear dose-dependence, with a requirement for prolonged pathway inhibition in order to significantly reduce tumor growth.

## CDK8/19 ligands have modest activity in patient-derived tumor xenograft models

To follow up our standard xenograft experiments, we next tested compound activity in human patient-derived tumor xenograft models (PDXs). Firstly, we determined if our compounds were active in in vitro soft agar clonogenic assays, using cells derived from 89 distinct PDX models from different tissue types. In this assay, colony growth was inhibited, but only by up to 50%, in one-third of PDX cell cultures treated with tool compound 2 (**Figure 3A–B**). Six of the more sensitive colorectal PDX tumor models were selected for in vivo monotherapy either with the lead compound 3 alone, or combined with a standard of care drug (irinotecan or oxaliplatin). Only two of these colorectal cancer PDX tumors models, CFX 883 and CFX 1753, responded to monotherapy with 3 (p<0.0001; **Figure 3—figure supplement 1A**). Importantly, the combination of 3 with standard of care therapies was only of statistically significant benefit in one colorectal tumor model, CFX 1034 (p<0.001; **Figure 3C** and **Figure 3—figure supplement 1**). Examination of p-STAT1$^{SER727}$ levels indicated near maximal CDK8 inhibition in the three tumors where p-STAT1$^{SER727}$ was tested (p<0.0001;

*Figure 3* and *Figure 3—figure supplement 1B and D*), indicating that the failure of 3 to slow PDX growth (*Figure 3—figure supplement 1C and F*) was unlikely to be due to a lack of target engagement.

The most CDK8/19 inhibitor sensitive cell model in the soft agar assays was one of the two acute myeloid leukaemia (AML) cell lines, Nomo-1, included in the cell panel. The Nomo-1 line was particularly sensitive to treatment with compound 2 (*Figure 3A*) with an 11 nM $GI_{50}$. Subsequently, in a Nomo-1 systemic in vivo model, treatment with 3 led to a potent reduction in circulating tumor cells (*Figure 3—figure supplement 2A–B*). An additional subcutaneous MV-4-11 AML xenograft also responded to monotherapy (TGI = 100%) with 10 mg/kg po qd 3, and also with 4 which showed evidence of near maximal target inhibition, as determined using p-STAT1$^{SER727}$ at 2 hr post-treatment (*Figure 3—figure supplement 2C–D*).

## Gene expression microarray profiling of human colorectal cancer xenografts post-treatment

Having demonstrated target engagement and associated antitumor activity in vivo, we further explored the molecular response of tumors treated with compounds 1 and 3 (*Figure 4—source data 1*). We initially identified 278 transcription factor-associated genesets that were enriched in genes whose expression was significantly altered following in vivo treatment of COLO205 or SW620 human tumor xenografts. The altered expression of 121/278 of these transcription factor-associated genesets was identified following treatment of both xenografts (*Figure 4A*). Of note we also identified 185 transcription factor-associated genesets that were significantly enriched in sets of genes associated with super-enhancers; of these, 2/3rds were shared with the genesets whose expression was altered by compound treatment (*Figure 4A*). These common transcription factor genesets included transcription factors known to be regulated by CDK8, such as TCF4, SMADs, STATs, c/EBP and HIF1A (*Figure 4—source data 1*) (*Bancerek et al., 2013*; *Morris et al., 2008*; *Zhao et al., 2013*; *Fryer et al., 2004*; *Alarcón et al., 2009*; *Zhao et al., 2012*). The genesets shared between both treatment and super-enhancers also encompassed transcription factors (NANOG, OCT3/4 and SOX2) required for stem cell pluripotency (*Figure 4—source data 1*) (*Whyte et al., 2013*).

A similar analysis, but this time of genes encoding specific pathway components, found 85 genesets were significantly enriched in the pool of genes whose expression was modulated by compound treatment (*Figure 4A*); 15 of which were found in both colorectal cancer xenograft experiments. A third of these were geneset-encoded gene products associated with WNT signaling, while others encoded components of individual pathways involved in development, inflammation or osteoblast biology (*Figure 4—source data 1*). Five of the genesets, common to both tumor models, all comprising WNT pathway components, were enriched with super-enhancer associated genes.

In addition, we used geneset enrichment analysis (GSEA) to compare the compound-treated profiles with sets of genes associated with super-enhancers identified from published ChIP-seq datasets (*Pelish et al., 2015*), and genesets from MSigDB (www.broadinstitute.org/msigdb). We found that treatment of both tumor models resulted in a consistent and significant modulation of genes in the vicinity of MED1 and CDK8-associated super-enhancers identified from ChIP-seq datasets (*Figure 4B–C* and *Figure 4—source data 1*). Interestingly, super-enhancers defined by a BRD4 ChIP-seq in lymphoid cells did not show the same significant modulation as the MED1/CDK8-defined super-enhancers (*Figure 4B–C*). The expression of many of the MSigDB genesets were positively or negatively correlated with the compound-treated samples and included genesets encoding gene products associated with, or regulated by, developmental, immunological or inflammatory pathways (*Figure 4—source data 1*). Some patterns were common to all treated samples, for example the genesets with decreased expression associated with stem cell pluripotency (SOX2, NANOG and OCT4) were modulated by treatment in both of the tumor models and also with both 1 and 3 (*Figure 4—figure supplement 1*). In contrast, other genesets exhibited cell line-specific regulation, for example the expression of bone morphogenic protein 2 (BMP2)-regulated genes and others associated with bone remodelling were inhibited by treatment of COLO205 xenografts, but were activated in SW620 xenografts (*Figure 4—figure supplement 1*).

Overall, the altered levels of transcripts from genes influenced by transcription factors or super-enhancers known to be regulated by the Mediator complex and CDK8 are consistent with our compounds affecting a broad range of CDK8/19-regulated gene transcription (*Mallinger et al., 2015*).

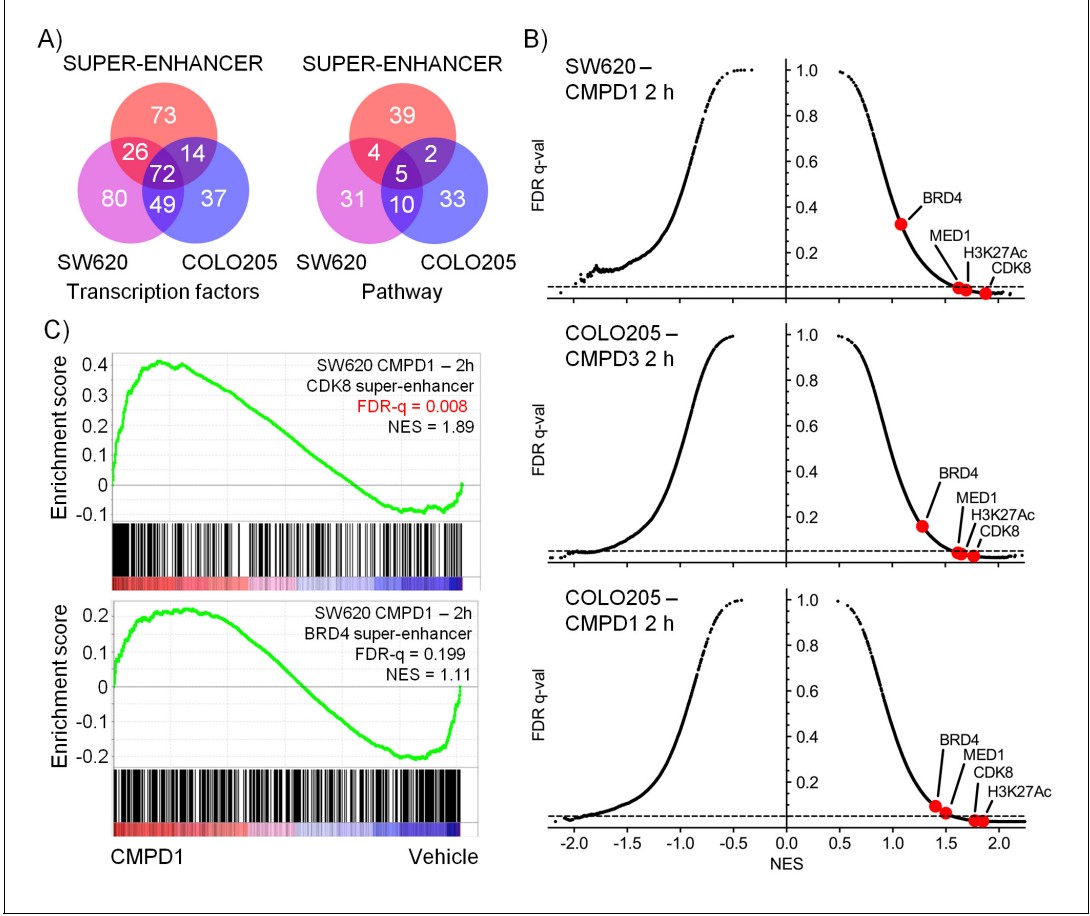

**Figure 4.** Microarray gene expression profiling following in vivo treatment of human colorectal cancer xenografts with CDK8/19 ligands. Mice were treated with 70 mg/kg po 1 (SW620 and COLO205), 20 mg/kg po 3 (COLO205). (**A**) Venn plots of transcription factor-associated genesets or those encoding or regulating pathways enriched in genes whose expression was significantly altered by treatment (Supplementary Dataset). (**B**) GSEA of CDK8 or BRD4-associated super-enhancer genes in treated human tumor xenografts. (**C**) Scatterplot of false discovery rate (FDR-q) versus normalized enrichment score (NES) for indicated gene sets evaluated by GSEA (n=10,218), signatures include those from MSigDB, dbSUPER and the ChIP-seq data from Pelish and colleagues (*Pelish et al., 2015*).

The following source data and figure supplement are available for figure 4:

**Source data 1.** Geneset expression analysis of microarray data following in vivo treatment of SW620 or COLO205 cells.

**Figure supplement 1.** Microarray gene expression profiling following in vivo treatment of colorectal cancer cell line xenografts.

The genes and genesets identified suggested that effects on stem cells, bone, immunology and inflammation might influence compound tolerability (*Figure 4—figure supplement 1*).

## Altered cell distribution in hyperplastic intestinal crypts following treatment

Among the profiles identified by GSEA were those associated with the regulation of normal and tumor stem cell populations (*Figure 4—source data 1*). We used a mouse model expressing doxycycline (Dox)-inducible activated beta-catenin (*Jardé et al., 2013*) to explore the effect of the tool compound **1** on an oncogenically-activated stem cell compartment. We separated the crypt-cell compartment from intestinal epithelial cells using a panel of cell surface markers (*Figure 5—figure supplement 1* and *Figure 5—source data 1*; *Wang et al., 2013*). GRP78 staining was then used to separate stem-like and transit amplifying-like (TA) cells, the success of which was confirmed by RT-PCR (*Figure 5A* and *Figure 5—source data 1*).

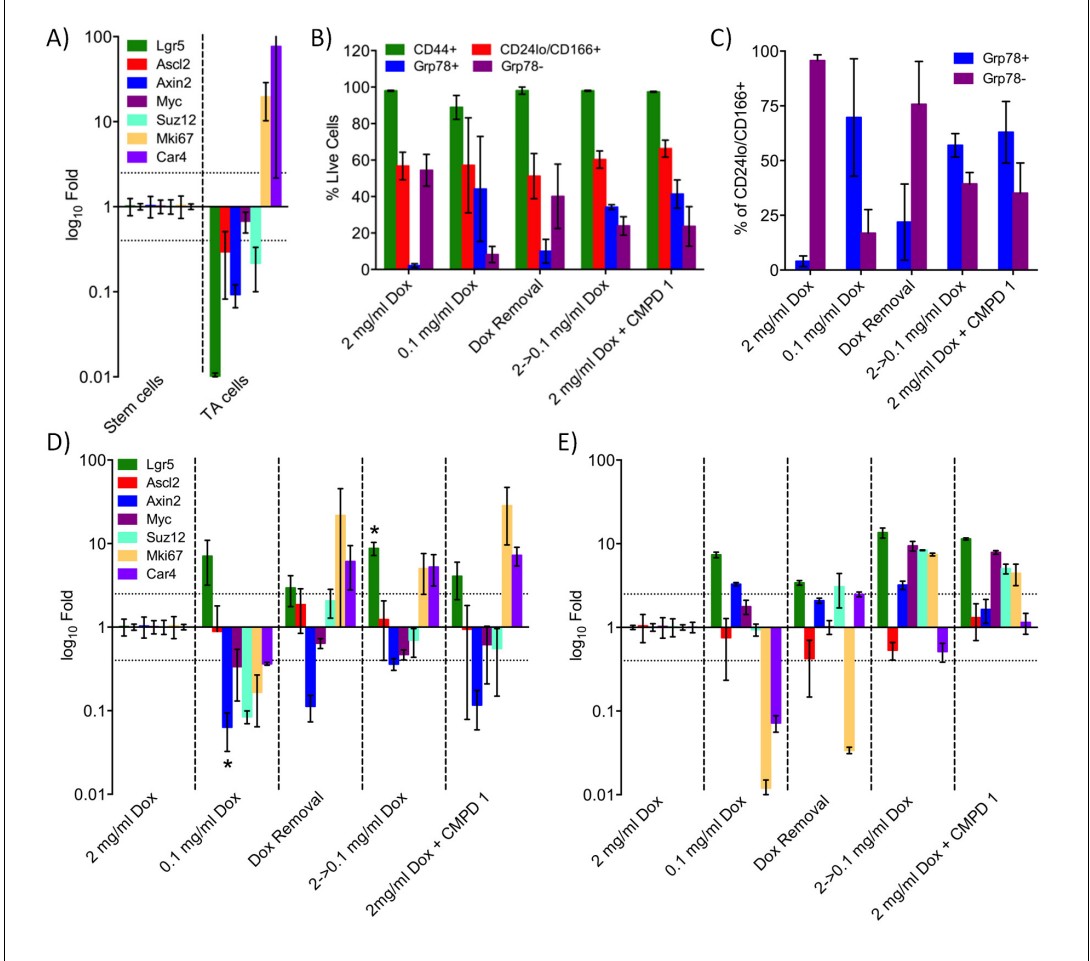

**Figure 5.** Treatment with CDK8/19 ligand 1 reduces the hyperplastic crypt stem cell population. Gene expression, measured by RT-PCR, in the intestinal epithelial stem and TA cells isolated from mice expressing a Dox-inducible activated β-catenin transgene. (A) Transcript abundance, relative to control, in stem and TA cells following induction with 2 mg/ml Dox. (B) Abundance of different cell types following treatment with Dox and compound 1. (C) Proportion of stem versus TA cells following treatment. (D) Fold changes in transcript abundance in stem (D) and TA (E) cells following treatment. All data are mean values ± s.d., n = 3.

The following source data and figure supplement are available for figure 5:

**Source data 1.** Antibodies and PCR primers used for analysis of mouse intestinal epithelial cells.

**Figure supplement 1.** Analysis of stem and TA cells isolated from the hyperplastic crypts of mice expressing a Dox-inducible activated β-catenin transgene.

High-level induction of beta-catenin with 2 mg/mL Dox significantly increased the percentage of stem cells (GRP78⁻) in the crypt cell compartment, whereas lower-level induction with 0.1 mg/mL Dox was associated with a larger transit amplifying (GRP78⁺) cell population, as previously described (*Figure 5B–C*) (*Hirata et al., 2013*). The complete withdrawal of Dox inhibited WNT signaling in the stem cell population, confirmed by a change in the abundance of transcripts from the WNT-regulated genes *Axin2* and *Car4* (*Figure 5D–E*). However, Dox withdrawal did not significantly alter the ratio of stem to TA cells (*Figure 5B–C*). In contrast, reducing the degree of Dox-induction from 2 to 0.1 mg/ml shifted the population towards an increased proportion of TA cells (*Figure 5B–C*). Consistent with the shift from stem to TA cell distribution, the gene expression pattern in the stem cell fraction was similar to Dox-removal, but was different from the TA cell population.

Previously, we demonstrated that treatment with >37.5 mg/kg compound 1 was sufficient to inhibit WNT signaling in this model (*Dale et al., 2015*). Here, we found that treatment with 1 at 75 mg/kg x 3 over 24 hr resulted in a gene expression pattern in the stem cell fraction that was similar to both Dox-removal and reduction of Dox from 2.0 to 0.1 mg/ml (*Figure 5D*). In contrast, the gene expression in the TA fraction following treatment with 1 was similar to the effect of reducing the level of Dox from 2.0 to 0.1 mg/ml, but was different from the Dox withdrawal condition (*Figure 5E*). This observation was consistent with the observation that treatment with 1 resulted in a shift in the population distribution from stem cell to TA similar to that seen by reducing the level of Dox from 2.0 to 0.1 mg/ml (*Figure 5B–C*). This implies that the inhibition of CDK8/19 by 1 reduces, rather than eliminates, WNT signaling in the oncogenically-activated stem cell compartment and it is this that alters the proportion of stem cells to proliferative TA cells in the hyperplastic crypt.

## Modulation of bone morphogenesis

GSEA also indicated that genes encoding products associated with the bone environment, such as genes regulated by BMP2, were preferentially affected by compound treatment. Having already determined the effect of 1 on the intestinal crypt stem cell population (*Figure 5*), we next investigated the effect of CDK8/19 inhibition on a bone progenitor cell model. We reasoned that, potentially, CDK8/19 inhibition could affect the ability of stem cells in the bone marrow to self-renew through inhibition of the WNT pathway or via a BMP-dependent signaling mechanism that requires SMAD-regulated transcription. SMAD is a transcriptional target of CDK8 (*Alarcón et al., 2009*) and both SMADs1-5 and BMP2-regulated gene expression were identified as significant genesets (*Figure 4—source data 1*).

We treated mouse KS483 osteoprogenitor cells with LGK974, a Porcupine inhibitor that inhibits WNT signaling, and observed reduced secretion of Procollagen type I N-terminal propeptide (PINP), an organic component of bone, and also reduced deposition of calcium, an inorganic component of bone (*Figure 6A–B*) (*Dang et al., 2002*). In contrast, compound 3 stimulated PINP secretion at low concentrations and inhibited PINP secretion at higher concentrations (*Figure 6A*), while calcium deposition was inhibited in a concentration-dependent manner (*Figure 6B*). These data indicate that our lead compound 3 adversely affects bone development in an in vitro model and that its effects are distinct from a specific inhibitor of WNT signaling.

## Modulation of immune and inflammatory response in co-culture models

Genesets associated with the immune and inflammatory response were selectively affected by our CDK8/19 ligands, prompting us to examine the effects of our compounds on immune and inflammatory responses (*Figure 4—source data 1*). In 12 single or co-culture in vitro tissue models we assayed 163 clinically relevant extracellular biomarkers including cytokines, chemokines, membrane receptors, matrix components, and proteases (*Figure 6—figure supplement 1* and *Figure 6—source data 1*) (*Berg et al., 2010*). All four compounds (1–4) elicited a similar biomarker response across the cell line panel (median r = 0.812; range: 0.630 – 0.924), indicating that the observed changes were the result of CDK8/19 inhibition rather than off-target effects. Comparing the effects of our compounds with a proprietary database of >3000 approved drugs and experimental agents failed to find any close matches (*Berg et al., 2010*). The levels of interleukin 17A and 17F usually rise and fall together (*Melton et al., 2013*), but treatment with our compounds elicited an unusual split response, with an increase in IL-17A and a corresponding decrease in IL-17F (BT condition: *Figure 6C*, *Figure 6—figure supplement 1* and *Figure 6—source data 1*). Overall, the pattern observed across the biomarkers was consistent with the effects of our compounds on inflammation (VCAM-1, sPGE2 and IL-8), immunomodulation (IL-17A, IL-17F, HLA-DR and IgG) and tissue remodelling (uPAR) (*Figure 6C* and *Figure 6—source data 1*). In additional studies in rats treated with 3 we measured the levels of ten plasma cytokines associated with the Th1/Th2 immune response. Of the cytokines measured, only IL-12, a proinflammatory and proimmunestimulatory cytokine, increased significantly (p<0.005, *Figure 6D*).

## Adverse effects on multiple organ systems

Compounds from both chemical series were sufficiently well tolerated in mice to enable antitumor experiments to be conducted with a dosage regimen that resulted in near-maximal inhibition of

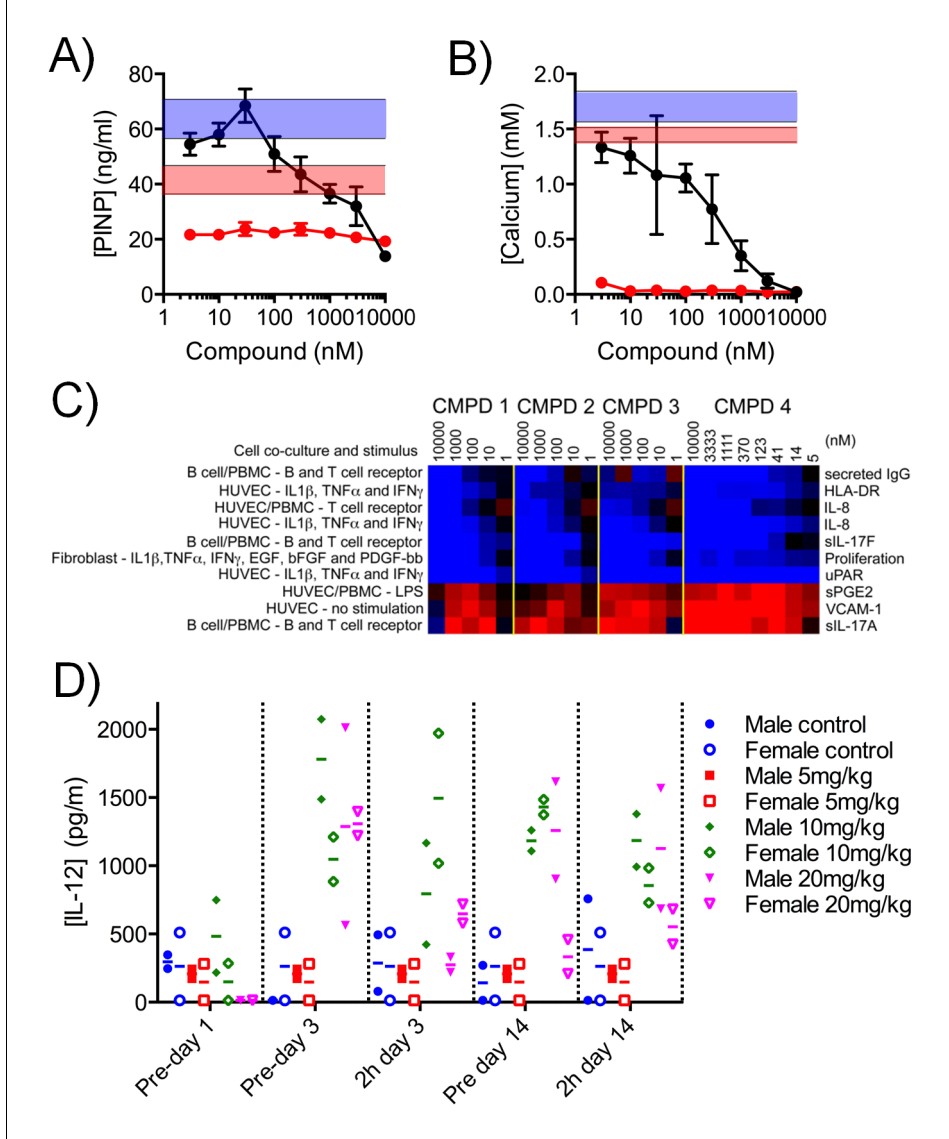

**Figure 6.** Effect of CDK8/19 chemical ligands on bone development and the immune response in model systems. Mouse KS483 osteoprogenitor cells were treated with LGK974 (red) or compound 3 (black) for 13 d and bone matrix formation determined by measuring (**A**) N-terminal propeptide of type I procollagen (PINP) and (**B**) calcium, in the external medium (mean ± s.d., n = 6). Blue region, level following 50 ng/ml BMP-2 (positive control); red region, basal level. (**C**) Heat map showing the 10 biomarkers most affected by compound treatment in cell co-culture models. Data are $\log_2$ ratios of biomarker levels following compound treatment relative to control (range: −0.874 to 0.396). Blue, decreased ratio; red, increased ratio. (**D**) Plasma IL-12 levels in Wistar rats treated with 5 – 20 mg/kg po qd 3. Two rats per cohort.

The following source data and figure supplement are available for figure 6:

**Source data 1.** Culture conditions and data from CDK8/19 ligand profiling in the culture/co-culture cell model panel.

**Figure supplement 1.** Effect of CDK8/19 ligands 1-4 on 12 culture/coculture cell models.

tumor STAT1$^{SER727}$ phosphorylation, showing that CDK8/19 activity was repressed for prolonged periods (*Figure 2*, *Figure 2—figure supplements 1–3*, *Figure 3*, *Figure 3—figure supplement 1*. However, there was some evidence for body weight loss that sometimes necessitated short dosing breaks (*Figure 2—source data 1*). Given this sporadic body weight loss in mice and our evidence

that the compounds had effects on bone development, altered immune/inflammatory profiles and stem cell differentiation in vitro or in vivo experimental models, we carried out detailed toleration studies. Our aim was to determine if there was a therapeutic window for the compounds, using PK/PD and efficacy data determined in our mouse models as a guide. Following daily doses of compound 3 or 4 for 14 days in rats we detected significant, adverse alterations in multiple organ systems and tissues (*Table 1*). In these toleration studies, we demonstrated extensive (80%) inhibition of STAT1$^{SER727}$ phosphorylation 6 hr post-treatment at all doses of 3 tested (*Figure 7—figure supplement 1*). The lowest doses administered resulted in plasma concentrations below or equivalent to the plasma exposures achieved at efficacious doses in our experimental human tumor xenograft efficacy models, suggesting the lack of a clear therapeutic window (*Table 1*). Consistent with our in vitro observations of the effects on bone maturation, the rat studies revealed two different, paradoxical effects on bone: an inhibitory effect resulting in dysplasia of the growth plate, a decrease in the proliferative zone and false endochondral ossification, and an activating effect resulting in proliferation of irregular woven bone in the bone cavity and below the periosteum (*Table 1* and *Figure 7*). Other adverse pathomorphological findings included necrotic and apoptotic cell lesions in the exocrine pancreas, gastrointestinal mucosa (stomach and duodenum), male reproductive tract (*Figure 7*), mammary gland, skin (hair follicle), heart (valvular interstitial cells) and lymphatic tissues (thymus, spleen, lymph nodes). Proliferative lesions were found in the lungs (bronchiolar epithelium and smooth muscle cells), liver (bile ducts and smooth muscle cells of hepatic arteries), thymus (epithelium free areas), mammary gland, male reproductive system and heart (valvular interstitial cells) (*Table 1*).

Follow-up studies in dogs indicated a similar, widespread adverse safety profile at therapeutically relevant exposures of 3 and 4 (*Table 1*). Since these pathological effects were seen with two highly

**Table 1.** CDK8/19 ligands 3 and 4 adversely affect multiple organs in rats and dogs.
Wistar rats (5 male and 5 female per cohort) or Beagle dogs (2 male and 2 female per cohort for 3 and 1 male and 1 female for 4) received a daily oral dose of 3 or 4 for 14 days. In the rat study of 4, all animals were prematurely culled at 60 mg/kg and one male and female at 20 mg/kg, as a result of compound toxicity. In the dog studies, all animals were prematurely culled in the study of 3 and one female following exposure to 4 as a result of toxicity. The most severely affected organs are indicated in bold. The fold efficacious dose was calculated from a plasma PK measurement of compound exposure in satellite animals run in parallel to the tolerability study and compared to exposures at efficacious doses in human tumour xenograft models in mice (m – male and f – female).

| | Rat | | | Dog |
|---|---|---|---|---|
| | Low dose | Mid dose | High dose | Low dose |
| CMPD 3 (mg/kg) | 5 | 10 | 20 | 5 |
| Target organs | Bone, **bone, marrow, heart, liver, lung,** lymph nodes, pancreas, reproductive tract (m), spleen, thymus. | Bone, bone marrow, heart, liver, lung, lymph nodes, pancreas, reproductive tract (m and f), spleen, thymus. | Bone, bone marrow, heart, liver, lung, lymph nodes, pancreas, reproductive tract (m and f), spleen, thymus. | Bone marrow, **gastrointestinal mucosa, heart, lymphatic system** |
| Fold of efficacious dose; 10 mg/kg | ~0.3 (m) − 1.3 (f) | ~0.5 (m) − 2 (f) | ~1 (m) − 5 (f) | ~0.3 (m) − 0.3 (f) |
| CMPD 4 (mg/kg) | 10 | 20 | 60 | 20 |
| Target organs | Bone, bone marrow, intestines, **liver,** lung, lymph nodes, **mammary gland,** pancreas, **reproductive tract (m and f), skin,** spleen, **stomach,** thymus. | Bone, bone marrow, heart, intestines, liver, lung, lymph nodes, **mammary gland,** pancreas, **reproductive tract (m and f), skin,** spleen, **stomach,** thymus | Bone, bone marrow, brain, heart, intestines, **liver, lung,** lymph nodes, **mammary gland,** pancreas, **reproductive tract (m and f), skin,** spleen, **stomach,** thymus | Bone marrow, **heart, Intestines,** lymphatic system |
| Fold of efficacious dose; 10 mg/kg 30 mg/kg | ~0.9 (m) − 2.4 (f) ~0.3 (m) − 0.8 (f) | ~3.9 (m) − 5.7 (f) ~1.3 (m) − 1.9 (f) | ~10.8 (m) − 23.1 (f) ~3.6 (m) − 7.7 (f) | ~22 (m) − 46 (f) ~7 (m) − 15 (f) |

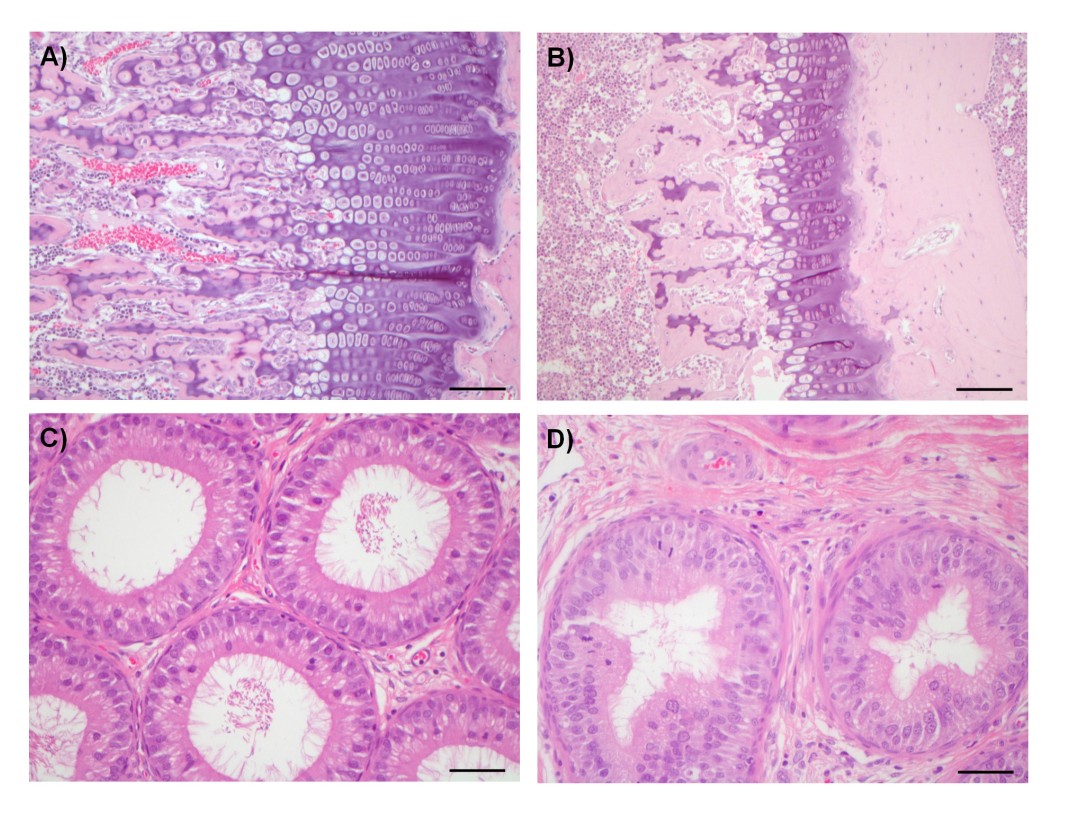

**Figure 7.** Examples of degenerative and proliferative lesions in rats treated with CDK8/19 ligands 3 or 4. (**A**) Intact proliferative zone in the bone growth plate of a control rat. (**B**) Dysplastic proliferative zone, showing disturbance of regular endochondrial ossification, from a rat treated with 20 mg/kg 3. Scale bar in **A** and **B** = 100 μm. (**C**) Intact epididymides, with epididymal cells, isolated from a control rat. (**D**) Epididymides with epithelial hyperplasia (distal corpus) isolated from a rat treated with 60 mg/kg 4. Scale bar in **C** and **D** = 50 μm.

The following figure supplement is available for figure 7:

**Figure supplement 1.** Pharmacodynamic response in Wistar rats treated with CDK8/19 ligand 3.

selective, but structurally distinct CDK8/19 inhibitors in both rats and dogs, we conclude that the adverse effects of treatment are the direct result of inhibition of CDK8 and/or CDK19. These observations indicate that the clinical development of either series of CDK8/19 inhibitors, or other chemotypes with similar profiles, would be extremely challenging.

## Discussion

CDK8 may reportedly act both as an oncogene and as a tumor suppressor, but until recently, the absence of a potent and selective inhibitor of CDK8 has restricted many functional studies to genetic inhibition using shRNA or siRNA (*Mitra et al., 2006*; *Chattopadhyay et al., 2010*; *Gu et al., 2013*; *Firestein et al., 2008*, *2010*; *Seo et al., 2010*; *Adler et al., 2012*; *Starr et al., 2009*). The potential difference between loss of protein and inhibition of enzymatic activity is highlighted by a recent study with the natural product CDK8/19 inhibitor cortistatin A that found a treatment-induced gene expression profile distinct from the profile resulting from CDK8/19 shRNA knockouts in the same cell line (*Poss et al., 2016*). Also conscious of the potentially opposing context-dependent roles of CDK8 in tumor development, we set out to clarify the therapeutic potential of targeting CDK8/19. Using two structurally-distinct series of potent and highly selective ligands that we discovered, we explored the consequences of CDK8/19 targeting in vitro and in vivo and investigated tolerability to determine if there was a therapeutic window (*Mallinger et al., 2015*, *2016a*; *Czodrowski et al.,*

2016). Despite reproducible inhibition of TCF/LEF reporter activity and STAT1$^{SER727}$ phosphorylation by both series of compounds, we were only able to detect modest, though generally significant, antiproliferative or antitumor effects in vitro or in vivo. In follow-up studies on cell cultures derived from PDX tumors, and selected PDXs in vivo, we found that compounds from the 3,4,5-trisubstituted pyridine series had only modest effects in vitro and little effect in vivo despite maximal inhibition of STAT1$^{SER727}$ phosphorylation. The pleiotropic nature of CDK8/19 function, influencing the activity of multiple specific transcription factors and also super-enhancers, may make identification of biomarkers for particular cancer cell types that are especially sensitive to CDK8/19 inhibitors challenging; despite this both series of compounds did show greater potency in a systemic or subcutaneous model of human AML, similar to that reported for the natural product inhibitor of CDK8/19, cortistatin A (*Pelish et al., 2015*).

We found that the in vivo activity of our CDK8/19 inhibitory compounds was associated with modulation of gene expression regulated by transcription factors that are CDK8 substrates. Moreover, Pelish and colleagues (*Pelish et al., 2015*) recently demonstrated that in AML cells CDK8 is associated with gene super-enhancers and that pharmacological inhibition of CDK8 activated superenhancer output. Here, we also found that pharmacological inhibition of CDK8 with our potent and selective chemical probes in colorectal cancer models resulted in gene expression profiles consistent with increased super-enhancer activity. Genes whose expression were altered by the compounds included those encoding products associated with bone development, stem cell biology, immunology and inflammation. Extensive follow up experiments in in vitro and in vivo models demonstrated effects on bone, stem cell differentiation and response of immune cells to different stimuli. Some of the effects were unique, for example concentration-dependent stimulation or inhibition of bone matrix production by osteoprogenitor cells in vitro that was not observed with a WNT-pathway inhibitor.

CDK8 activity maintains embryonic stem cells in an undifferentiated, pluripotent state and colorectal tumors in a de-differentiated state (*Adler et al., 2012*). We found that complete inhibition of CDK8/19 in the presence of activated oncogenic beta-catenin mimicked the effect of reducing, rather than completely abrogating, WNT-signaling. CDK8/19 inhibition resulted in a shift from a stem cell to a predominantly TA cell phenotype. This response may also be linked to super-enhancer activation, as expression of *Myc*, a super-enhancer-regulated gene (*Lovén et al., 2013*), was repressed in the stem cell population, but elevated in the TA cell population following compound treatment. The example of *Myc* illustrates the potential complexity of the response to CDK8 inhibition, as *Myc* expression will potentially be repressed through loss of CDK8 activity required by betacatenin, but *Myc* expression may also be promoted through super-enhancer activation. This suggests that cellular context will have a major impact on the transcriptional response to CDK8 inhibition.

The final key aim of our studies was to investigate tolerability to CDK8/19 inhibition and identify a possible potential therapeutic window for compounds 3 and 4. In rat and dog tolerability studies, we found that 3 and 4 produced unusually extensive, but similar, adverse effects in a wide range of tissues. Given our observations, that are consistent with the reported role of CDK8 in repressing super-enhancer activity (*Pelish et al., 2015*), and the potentially key roles of super-enhancers as master controllers of cell identity and function (*Whyte et al., 2013*), the breadth, degree and depth of adverse effects is perhaps not surprising. We detected elevation of the mainly proinflammatory and proimmunestimulatory associated cytokine, IL-12, in rats treated with 3. Elevated IL-12 may result from decreased STAT1$^{SER727}$ following CDK8 inhibition, as bone marrow macrophages from STAT1$^{S727A}$ mutant mice exhibit elevated IL-12 and Cox-2 expression following activation (*Schroder et al., 2007*). Cox-2 induction is associated with an increased production of PGE2, which we detected in vitro, that may also explain the unusual split response of increased IL-17A and decreased IL-17F observed in our in vitro experiments, since PGE2, or agonists of the EP receptor for PGE2 are among the few stimuli that induce this split effect (*Melton et al., 2013*).

We concluded that the multiplicity of preclinical pathomorphological lesions would make monitoring and controlling toxicity in a clinical study very challenging, especially as no clear safety window could be established from our studies. Our data strongly suggest that the adverse effects are target related as they are detected with two chemically distinct series of potent and selective CDK8/19 ligands and indicate that dual pharmacological CDK8/19 inhibition is not tolerated in rats or dogs at exposure levels that correspond to therapeutically relevant exposures. Even in the more sensitive xenograft models of AML, there was no clear therapeutic window, as these tumors responded only

when STAT1$^{SER727}$ phosphorylation was continuously inhibited and equivalent exposures were not tolerated in our *in vivo* rat and dog toleration studies. Protein kinase inhibitors in clinical development or approved often have off-target activity or in some cases can be intentionally multi-targeted to have inhibitory activity against multiple protein kinases. Our observations are important for both scenarios as we identify CDK8/19 as potential 'anti-targets' to be avoided and we recommend screening against these protein kinases when establishing the safety profile of lead compounds and development candidates as well as assessing the quality of chemical probes.

As described earlier, the natural product cortistatin A was recently reported by Pelish and colleagues as a CDK8/19 inhibitor with specificity, potency, favourable pharmacokinetics that would make it a useful in vitro and in vivo probe for the Mediator kinases and as a promising lead for development of therapeutics (*Pelish et al., 2015*). Here, we employed two different series of potent and selective CDK8/19 inhibitors, with paired negative controls, that are much less challenging to synthesise compared to cortistatin A (*Pelish et al., 2015*). Our synthetic compounds have optimal pharmacological and pharmaceutical properties with single digit nM affinities for CDK8/19 and low nanomolar activity against promoter and STAT1$^{SER727}$ reporter assays and are as potent as cortistatin A, as well as exhibiting very high selectivity in broad kinome profiling. As described by Pelish and colleagues for cortistatin A, we found similar induction of super-enhancer-regulated genes with our compounds, although note that in our case we profiled gene expression in tumours treated in vivo rather than in vitro (*Pelish et al., 2015*). Similar to cortistatin A our compounds were also tolerated in mice and likewise we also found AMLs to be highly sensitive in vivo models (*Pelish et al., 2015*). However, in our study we also evaluated our two series in dedicated tolerability studies in rat and dog that revealed toxicity not apparent in mouse studies. The detailed toxicity profile of cortistatin A was not reported by Pelish and colleagues, but given the similarities in the results from in vitro and in vivo studies between cortistatin A and our two series we predict that cortistatin A and other CDK8/19 inhibitors would exhibit similar toxicity.

It remains to be seen if toxicity could be avoided if CDK8/19 inhibitors were administered intermittently as part of a combination therapy. For example, CDK8/19 inhibitors might modulate antitumor immunotherapy by inactivating STAT1 and stimulating tumor surveillance by NK cells (*Putz et al., 2013*). It is also unclear if toxicity could be avoided by selectively targeting CDK8 or CDK19 alone. During the optimisation of both the 3,4,5-trisubstituted pyridine and 3-methyl-1*H*-pyrazolo[3,4-*b*]pyridine series, we were unable to separate CDK8 from CDK19 affinity (*Mallinger et al., 2015*, *2016a*; *Czodrowski et al., 2016*). This is also true for two additional distinct chemical series that we identified (*Schiemann et al., 2016*; *Mallinger et al., 2016b*), and also for a further three chemotypes we profiled from the literature, all of which demonstrated balanced CDK8 and CDK19 affinity in our hands. Similarly, the natural product cortistatin A cannot selectively distinguish between CDK8 and 19 (*Pelish et al., 2015*). The inability to selectively target CDK8 or CDK19 is likely due to the very high degree of sequence similarity around the active site of these two kinases (*Figure 1c*), suggesting that strategies to selectively target either of these paralogs will require a different approach, such as allosteric modulation or selective degradation.

The recent identification of selective inhibitors of the transcriptional kinases CDK7 or CDK12/13 has fuelled interest in the clinical development of inhibitors of these targets (*Kwiatkowski et al., 2014*; *Wang et al., 2015*; *Zhang et al., 2016*; *Christensen et al., 2014*; *Chipumuro et al., 2014*). This is of relevance here as inhibition of these kinases has also been reported to affect super-enhancer-associated gene expression in T cell acute lymphoblastic leukaemia, B cell chronic lymphocytic leukaemia, MYCN-driven tumours, small cell lung cancer and triple negative breast cancer models where anti-tumour activity is observed (*Kwiatkowski et al., 2014*; *Wang et al., 2015*; *Zhang et al., 2016*; *Christensen et al., 2014*; *Chipumuro et al., 2014*). In adult CDK7 conditional knockout mice, effects were seen in tissues with a high cell turnover that were at least in part due to a depleted stem cell population resulting from a loss of CDK1 and CDK2 activation or reduced super-enhancer-associated gene expression (*Ganuza et al., 2012*). Super-enhancers are frequently found associated with genes whose products control the pluripotent state or define cell identity and this may make stem cell populations particularly vulnerable to therapeutic interventions that interfere with super-enhancer associated gene expression, including inhibitors of the transcription-regulating CDKs (*Whyte et al., 2013*). In an in vivo mouse model of an oncogenically-activated stem cell compartment we found our CDK8/19 inhibitors altered the proportion of stem cells to proliferative TA cells that may in part be due to super-enhancer activation. This raises the hypothetical possibility

that, similar to AML cells (*Pelish et al., 2015*), the stem cell compartment requires a precise 'dosage' of super-enhancer activity and that activation or inhibition of super-enhancer activity will negatively impact the stem cell compartment. The selective CDK7 tool inhibitors are reported to be tolerated in mice and so predicted to be non-toxic (*Kwiatkowski et al., 2014*; *Wang et al., 2015*; *Christensen et al., 2014*; *Chipumuro et al., 2014*); however, our CDK8/19 inhibitors were also tolerated in mice and the toxicity was not revealed until detailed tolerability studies were performed in other species. Thus the true toxicity profile of other transcriptional CDK inhibitors may not be revealed until dedicated tolerability studies are performed using selective compounds with optimised pharmaceutical properties.

In summary, we have discovered and made available two chemically distinct series of potent selective chemical probes and appropriate inactive control compounds that can be used to further explore the function of Mediator-associated kinases CDK8/19 and their role in human disease both in vitro and in vivo. These compounds will also be of particular value for exploring the regulation of super-enhancer activity in development and disease. However, on the basis of the complex toxicological profile and an inability to define a clear therapeutic window, we have decided against the further clinical development of our compounds and suggest caution when considering the clinical applicability of other CDK8/19 inhibitors. We also advise incorporating the profiling CDK8/19 as anti-targets in drug discovery and chemical probe projects aimed at other kinases.

## Materials and methods

### Compounds

Compounds were prepared as described (*Mallinger et al., 2015*, *2016a*; *Czodrowski et al., 2016*) or resynthesised by published routes.

### Biochemical assays

In vitro binding of compounds to CDK8 was determined using FRET-based Lanthascreen binding competition with a dye-labeled ATP competitive tracer assay. Alternatively, we used a reporter displacement assay provided by Proteros Biostructures GmbH (Germany) for CDK8 or CDK19 as described previously (*Mallinger et al., 2016a*). The human CDK8–CCNC complex was expressed, purified and crystallized as described previously (*Dale et al., 2015*). Crystals were back-soaked for different times and concentrations of ligand before being selected for structure determination (*Dale et al., 2015*).

### Cell culture

Only authenticated and mycoplasma-free cell lines were used in this study. All cancer cell lines used in this study (COLO205 - RRID:CVCL_0218; DLD1 - RRID:CVCL_0248; HT29 - RRID:CVCL_0320; LS174T - RRID:CVCL_1384; LS180 - RRID:CVCL_0397; LS513 - RRID:CVCL_1386; RKO - RRID:CVCL_0504; SW620 - RRID:CVCL_0547; SW837 - RRID:CVCL_1729; SW948 - RRID:CVCL_0632) were obtained from the ATCC (LGC Promochem, UK), were regularly tested and confirmed as mycoplasma-free (Lonza, UK) and were authenticated by short tandem repeat (STR) analysis profiling. Multiplex amplification of genomic loci Penta E, D18S51, D21S11, TH01, D3S1358, FGA, TPOX, D8S1179, vWA, Amelogenin, Penta D, CSF1PO, D16S539, D7S820, D13S317, and D5S818 is performed using a PowerPlex 16 HS (Promega, Madison, Wisconsin, USA). STR sequences were analyzed on an Applied Biosystems 3500xL Genetic Analyzer (Thermo Fisher Scientific Life Technologies, UK) and compared to different cell line reference databases. Soft agar PDX cell cultures were run at Oncotest (Germany). Cell lines transduced with a TCF/LEF fLUC reporter were generated and assayed as described previously (*Mallinger et al., 2015*; *Dale et al., 2015*). Inducible shRNA knockout models were established in the Colo205-F1921 subline carrying the TCF/LEF fLUC reporter have been described previously (*Dale et al., 2015*).

HT29 cells ($5 \times 10^4$/ml) were reverse transfected in 6 well plates with 3.75 μl/ml LipofectamineR-NAiMAX (ThermoFisher Scientific, UK) in a 2 ml final volume with 50 nM pooled siRNA targeting CDK8, CDK19 or a non-silencing control siRNA (Quiagen cat. GS1024, GS23097 and 1027280 respectively). Viability was determined by resazurin staining (R&D systems, UK) for 30 min 5 d post-transfection.

## Bone culture assays

Bone formation assays were conducted by Pharmatest (Finland) in mouse KS483 osteoprogenitor cells cultured for 13 d with LGK974 or 3. The amount of PINP secreted into the culture medium, and calcium deposition, were measured by ELISA (Roche Diagnostics, UK).

## In vitro coculture studies

Seven different primary cells types were exposed to different stimuli as single or co-cultures and the response of relevant extracellular biomarkers assayed. Biomarker assays were run at DiscoveRx (Fremont, California, USA) (*Figure 6—source data 1*). Cultures were treated with DMSO (vehicle) or a dilution series of 1 – 10000 nM of test compounds. Profiles were compared to a proprietary database of compounds previously profiled in this system (*Berg et al., 2010*).

## Animal studies

In the UK, all animal work was conducted in accordance with the National Institute for Cancer Research guidelines (*Workman et al., 2010*), with the research programme and procedures approved by the local Animal Welfare and Ethical Review Boards and subject to UK Government Home Office regulations (Licence PPL 70/7635 & PPL 30/3279). In Germany the animal work was carried out in accordance with the German Law on the Protection of Animals (Article 8a) and the pertaining files at the local animal welfare authorities in Darmstadt and Freiburg bear the references DA/375, DA4/1003, DA4/1004 and G13/13 respectively. The studies were designed in accordance with presently valid international study guidelines (e.g. ICH guideline M3 R2) and performed in compliance with animal health and welfare guidelines.

### Tumor xenograft studies

The establishment and treatment of cell line xenografts was performed as described previously (*Mallinger et al., 2015*; *Dale et al., 2015*; *Mallinger et al., 2016a*). PDX experiments were conducted at Oncotest (Germany).

### Pharmacokinetic and tolerability studies

In vitro intrinsic clearance was calculated from the rate of the compound disappearance using mouse, rat or human microsomes. The apparent permeability coefficients were calculated for compounds using Caco-2 monolayers seeded on polycarbonate filters as described previously (*Mallinger et al., 2015*, *2016a*).

In vivo live phase PK of compounds was determined in female NMRI mice, male Wistar rats or female beagle dogs receiving the compound either as a single intravenous (bolus) injection or an oral administration (by gavage) of the compound. The concentrations of compound were quantified using an UPLC method with tandem mass spectrometric detection (*Mallinger et al., 2015*, *2016a*). The tolerability studies were designed to meet national and international regulatory requirements for the conduct of short-term studies in animals. In brief, Wistar rats or beagle dogs were treated with a daily oral dose of the vehicle or compound for 14 days. Animals were culled 24 hr after the last dose. Additional satellite animals for toxicokinetic evaluations were included. Investigations included, but were not limited to: clinical signs, mortality, body weight, clinical pathology, cytokine measurements and histopathology.

The ICR does not use non-rodent species in research and, where this is deemed essential, requires ethical approval for use by organizations with whom we collaborate. Pharmacokinetic and tolerability analysis of compounds 3 and 4 in dogs, necessary for prediction of human pharmacokinetics, was approved by the ICR Animal Welfare and Ethical Review Board. Studies were sponsored and conducted in full compliance with national regulations at an Association for Assessment and Accreditation of Laboratory Animal Care accredited site of Merck Biopharma.

### In vivo stem cell analysis

Crypt structures were isolated from inducible Tet-O-$\beta$-catenin mice following treatment regimes described in *Figure 5—figure supplement 1* (*Jardé et al., 2013*). Mice were orally gavaged with 75 mg/kg 1 × 3 over 24 hr and doxycycline in drinking water supplemented with an i.p. dose of 0.1 mg/ml or 2.0 mg/ml doxycycline in PBS at time of gavage to maintain levels. Isolated crypts were

dissociated into single cells, then resuspended in staining medium containing DAPI and the antibodies shown in *Figure 5—source data 1*. Cells were sorted into lysis buffer, and gene expression determined by RT-PCR using the primers shown in *Figure 5—source data 1*

## Gene expression microarray profiling

Total RNA was extracted from xenograft tumors using a MagNa Pure 96 high-throughput robotic workstation (Roche Diagnostics, UK) and analyzed by microarray expression profiling (*Dale et al., 2015*). Purified, labeled cDNA products were hybridized to $8 \times 60K$ human microarrays (Agilent) and analyzed using Genespring (Agilent Santa Clara, California, USA, RRID:SCR_009196). Significantly differentially expressed genes were investigated for enrichment in terms of particular pathways or potential transcription factor regulation using the Metacore software (Thomson Reuters, New York City, New York, USA, RRID:SCR_008125). Lists of super-enhancer-associated genes were derived from published ChIP-seq datasets (dbSUPER; http://bioinfo.au.tsinghua.edu.cn/dbsuper/ and *Pelish et al., 2015*). GSEA was performed on the super-enhancer gene lists and also genesets from MSigDB (www.broadinstitute.org/msigdb). The links for the signatures used for the GSEA software are available in the Datasets section. Users are required to register to view the MSigDB gene sets and/or download the GSEA software. Microarray data are available on the NCBI Gene Expression Omnibus (GEO; http://www.ncbi.nlm.nih.gov/geo/) website under accession number GSE80472.

## Measurement of CDK8, CDK19, p-STAT1$^{SER737}$ and total-STAT1 protein

Levels of p-STAT1$^{SER727}$ and total STAT1 were quantified from cell or tumor lysates by immunoblotting, luminex or electrochemiluminescent ELISA as previously described in detail in (*Mallinger et al., 2015*; *Dale et al., 2015*; *Mallinger et al., 2016a*; *Czodrowski et al., 2016*). Proteins were also detected using an automated capillary immunoassay system (Protein Simple) with antibodies specific for CDK8 (Cell Signaling, Danvers Massachusetts, USA #4106, RRID:AB_1903936), CDK19 (Sigma-Aldrich, UK, HPA007053, RRID:AB_1233803) phospho-STAT1$^{SER727}$ (Cell Signaling, #8826), total STAT1 (Santa-Cruz Biotechnology, Dallas, Texas, USA, #346, RRID:AB_632435), and B-actin (Cell Signalling, #4970, RRID:AB_2223172), subsequently immunodetected using a horseradish peroxidase (HRP)-conjugated secondary antibody and chemiluminescent substrate.

## Cytokine measurement

Lithium-heparin-plasma (120 µL) was taken from each non-fasted animal on day 1 pre-dose and on days 3 and 14 pre-dose and also 2 hr after the last treatment. The rat Th1/Th2 multiPlex bead immunoassay panel (Invitrogen) was used to generate calibration curves and to measure cytokine (IL-1$\alpha$, IL-1$\beta$, IL-2, IL-4, IL-6, IL-10, IL-12, IFN$\gamma$, gmCSF and TNF$\alpha$) levels (Luminex, Austin, Texas, USA).

## Acknowledgements

This work was supported by Cancer Research UK (grant numbers C309/A11566, C368/A6743 and A368/A7990). We acknowledge Cancer Research UK funding to the Cancer Research UK Cancer Therapeutics Unit at The Institute of Cancer Research (ICR). We also acknowledge Cancer Research UK funding to the Cancer Research UK Centre at the ICR and Royal Marsden Hospital, and The National Health Service (NHS) funding to the National Institute for Health Research (NIHR) Biomedical Research Centre at the same institutions. We acknowledge funding by Breast Cancer Now (grant number 2008 MayPR16). Paul Workman is a Cancer Research UK Life Fellow. We thank Stefanie Gaus for expert technical assistance and Nicky Evans for editorial assistance.

## Additional information

### Competing interests

PAC, M-JO-R, RT, OA-P, GB, WC, SG, ADHB, SMH, AM, FRa, TR, MV, JB, SAE, PW: Current or former employee of The Institute of Cancer Research, which has a commercial interest in the development of WNT pathway inhibitors. SC, SEB, CE, PH, WK, FRo, KS, SS, RS, SW, AB, SH, KU, DW:

Current or former employee of Merck KGaA (Darmstadt, Germany), which has a commercial interest in the development of WNT pathway inhibitors. The other authors declare that no competing interests exist.

## Funding

| Funder | Grant reference number | Author |
|---|---|---|
| Breast Cancer Now | 2008MayPR16 | Trevor C Dale |
| Cancer Research UK | C309/A11566 | Paul Workman |
| Cancer Research UK | C368/A6743 | Paul Workman |
| Cancer Research UK | A368/A7990 | Paul Workman |

The funders had no role in study design, data collection and interpretation, or the decision to submit the work for publication.

## Author contributions

PAC, FRa, KS, RS, JB, TCD, SAE, DW, Conception and design, Analysis and interpretation of data, Drafting or revising the article; M-JO-R, RT, OA-P, GB, WC, SC, KE, SG, ADHB, PH, TR, MV, SW, Acquisition of data, Analysis and interpretation of data, Drafting or revising the article; SEB, WK, Analysis and interpretation of data, Drafting or revising the article; CE, AM, FRo, SS, Conception and design, Acquisition of data, Analysis and interpretation of data, Drafting or revising the article; SMH, Conception and design, Acquisition of data, Drafting or revising the article; AB, SH, KU, PW, Conception and design, Drafting or revising the article

## Author ORCIDs

Paul A Clarke, http://orcid.org/0000-0001-9342-1290
Julian Blagg, http://orcid.org/0000-0002-7409-0323
Paul Workman, http://orcid.org/0000-0003-1659-3034

## Ethics

Animal experimentation: In the UK, all animal work was conducted in accordance with National Institute for Cancer Research guidelines, with the research programme and procedures approved by the local Animal Welfare and Ethical Review Boards and subject to UK Government Home Office regulations (Licence PPL 70/7635 & PPL 30/3279). In Germany the animal work was carried out in accordance with the German Law on the Protection of Animals (Article 8a) and the pertaining files at the at the local animal welfare authorities in Darmstadt and Freiburg bear the references DA/375, DA4/1003, DA4/1004 and G13/13 respectively. The studies were designed in accordance with presently valid international study guidelines (e.g. ICH guideline M3 R2) and performed in compliance with animal health and welfare guidelines. The Institute of Cancer Research does not use non-rodent species in research and, where this is deemed essential, requires ethical approval for use by organizations with whom we collaborate. Pharmacokinetic and tolerability analysis of compounds in dogs, necessary for prediction of human pharmacokinetics, was approved by the ICR Animal Welfare and Ethical Review Board. Studies were sponsored and conducted in full compliance with national regulations at an Association for Assessment and Accreditation of Laboratory Animal Care accredited site of Merck Biopharma.

# Additional files

## Major datasets

The following dataset was generated:

| Author(s) | Year | Dataset title | Dataset URL | Database, license, and accessibility information |
|---|---|---|---|---|
| Clarke PA, Ortiz-Ruiz M-J, TePoele R, Adeniji-Popoola O, Box G, Court W, El Bawab S, Esdar C, Ewan K, Gowan S, de Haven Brandon A, Hewitt P, Hobbs SM, Kaufmann W, Mallinger A, Raynaud F, Roe T, Rohdich F, Schiemann K, Simon S, Schneider R, Valenti M, Blagg J, Blaukat A, Dale T, Eccles SA, Hecht S, Urbahns K, Workman P, Wienke D | 2016 | Assessing the mechanism and therapeutic potential of modulators of the human mediator complex-associated protein kinases | https://www.ncbi.nlm.nih.gov/geo/query/acc.cgi?acc=GSE80472 | Publicly available at the NCBI Gene Expression Omnibus (accession no: GSE80472) |

The following previously published datasets were used:

| Author(s) | Year | Dataset title | Dataset URL | Database, license, and accessibility information |
|---|---|---|---|---|
| Pelish HE, Liau BB, Nitulescu II, Lemieux ME, Shair MD | 2015 | Effect of cortistatin A (CA) on enhancer occupancy in CA-sensitive and -insensitive human cell lines | https://www.ncbi.nlm.nih.gov/geo/query/acc.cgi?acc=GSE65138 | Publicly available at the NCBI Gene Expression Omnibus (accession no: GSE65138) |
| Subramanian A, Tamayo P, Mootha VK, Mukherjee S, Ebert BL, Gillette MA, Paulovich A, Pomeroy SL, Golub TR, Lander ES, Mesirov JP | 2007 | Molecular Signatures Database v5.1 | http://software.broadinstitute.org/gsea/msigdb/download_file.jsp?filePath=/resources/msigdb/5.1/c2.all.v5.1.symbols.gmt | Available at the Gene Set Enrichment Analysis site (http://software.broadinstitute.org/gsea/msigdb/). Users are required to register to view the MSigDB gene sets and/or download the GSEA software |
| Subramanian A, Tamayo P, Mootha VK, Mukherjee S, Ebert BL, Gillette MA, Paulovich A, Pomeroy SL, Golub TR, Lander ES, Mesirov JP | 2007 | Molecular Signatures Database v5.1 | http://software.broadinstitute.org/gsea/msigdb/download_file.jsp?filePath=/resources/msigdb/5.1/c7.all.v5.1.symbols.gmt | Available at the Gene Set Enrichment Analysis site (http://software.broadinstitute.org/gsea/msigdb/). Users are required to register to view the MSigDB gene sets and/or download the GSEA software |
| Subramanian A, Tamayo P, Mootha VK, Mukherjee S, Ebert BL, Gillette MA, Paulovich A, Pomeroy SL, Golub TR, Lander ES, Mesirov JP | 2007 | Molecular Signatures Database v5.1 | http://software.broadinstitute.org/gsea/msigdb/download_file.jsp?filePath=/resources/msigdb/5.1/c5.bp.v5.1.symbols.gmt | Available at the Gene Set Enrichment Analysis site (http://software.broadinstitute.org/gsea/msigdb/). Users are required to register to view the MSigDB gene sets and/or download the GSEA software |

| Subramanian A, Tamayo P, Mootha VK, Mukherjee S, Ebert BL, Gillette MA, Paulovich A, Pomeroy SL, Golub TR, Lander ES, Mesirov JP | 2007 | Molecular Signatures Database v5.1 | http://software.broadinstitute.org/gsea/msigdb/download_file.jsp?filePath=/resources/msigdb/5.1/c3.tft.v5.1.symbols.gmt | Available at the Gene Set Enrichment Analysis site (http://software.broadinstitute.org/gsea/msigdb/). Users are required to register to view the MSigDB gene sets and/or download the GSEA software |
|---|---|---|---|---|
| Subramanian A, Tamayo P, Mootha VK, Mukherjee S, Ebert BL, Gillette MA, Paulovich A, Pomeroy SL, Golub TR, Lander ES, Mesirov JP | 2007 | Molecular Signatures Database v5.1 | http://software.broadinstitute.org/gsea/msigdb/download_file.jsp?filePath=/resources/msigdb/5.1/c3.tft.v5.1.symbols.gmt | Available at the Gene Set Enrichment Analysis site (http://software.broadinstitute.org/gsea/msigdb/). Users are required to register to view the MSigDB gene sets and/or download the GSEA software |
| Subramanian A, Tamayo P, Mootha VK, Mukherjee S, Ebert BL, Gillette MA, Paulovich A, Pomeroy SL, Golub TR, Lander ES, Mesirov JP | 2007 | Molecular Signatures Database v5.1 | http://software.broadinstitute.org/gsea/msigdb/download_file.jsp?filePath=/resources/msigdb/5.1/h.all.v5.1.symbols.gmt | Available at the Gene Set Enrichment Analysis site (http://software.broadinstitute.org/gsea/msigdb/). Users are required to register to view the MSigDB gene sets and/or download the GSEA software |
| Subramanian A, Tamayo P, Mootha VK, Mukherjee S, Ebert BL, Gillette MA, Paulovich A, Pomeroy SL, Golub TR, Lander ES, Mesirov JP | 2007 | Molecular Signatures Database v5.1 | http://software.broadinstitute.org/gsea/msigdb/download_file.jsp?filePath=/resources/msigdb/5.1/c6.all.v5.1.symbols.gmt | Available at the Gene Set Enrichment Analysis site (http://software.broadinstitute.org/gsea/msigdb/). Users are required to register to view the MSigDB gene sets and/or download the GSEA software |

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
