## [Decision Letter]

Thank you for submitting your article "Assessing the mechanism and therapeutic potential of modulators of the human mediator complex-associated protein kinases" for consideration by *eLife*. Your article has been reviewed by two peer reviewers, and the evaluation has been overseen by a Reviewing Editor and Charles Sawyers as the Senior Editor. The following individual involved in review of your submission has agreed to reveal his identity: Nathanael Schiander Gray (Reviewer #1).

The reviewers have discussed the reviews with one another and the Reviewing Editor has drafted this decision to help you prepare a revised submission.

Both reviewers and the editors found your manuscript to be important. The discoveries related to the pharmacological effects of selective Cdk8/19 inhibitors, including their potentially limited therapeutic window, should be of broad interest to the scientific community. The reviewers have recommended some minor revisions to improve the clarity of the manuscript. Please read their comments and revise your manuscript accordingly. As part of this process, we would also like for you to shorten the Impact statement. We do not anticipate that the revised manuscript will require re-review before acceptance.

Reviewer #1:

This is an amazing piece of work that deserves to be published quickly. An extraordinary amount of work has gone into these studies which appear to have been performed in a rigorous fashion. The conclusions are well supported by the data provided. Normally a study of this magnitude could only afford to be performed in a pharmaceutical company who would then not publish the work, especially the toxicology findings. I see the impact of this work being 1) clear therapeutic- implications for efforts to target Cdk8/19, 2) release of valuable chemical probes, 3) demonstration of the context-dependent effects of Cdk8/19 inhibition (most studies would just look at one context and then conclude target does x).

On the negative side, this manuscript is a dense read and does come across as a data-dump of sorts but I don't see an easy way to prune given the enormous volume of data.

*Reviewer #2:*

Clarke et al. describe the "de-validation" of CDK8/19 as suitable targets in both solid and liquid tumor indications. A great deal of interest in targeting super-enhancers in oncology has emerged from the work of Young, Gray, Shair, and others including Clarke et al. The natural product corticostatin A which targets CDK8/19 highly selectively was shown by Shair (Nature, 2015) to activate super enhancer gene sets which include tumor suppressive genes. The Clarke et al. team previously (Nature Chem. Bio., 2015) came across CDK8/19 as the relevant target in a hit from a Wnt-cell based screen. Thus, having identified a new chemical series for inhibition of CDK8/19, they asked whether using a panel of different structurally distinct CDK8/19 inhibitors could be employed to further test CDK8/19 as a target in oncology. In short, through careful pathway readouts (Stat phosphorylation) and cell killing assays in tumor cell lines as well as patient derived xenografts, they show that inhibition of CDK8/19 and by extension, activation of mediator, has complex effects in multiple organ systems, bone and immune cells in both ex vivo and *in vivo* settings. The most sensitive cancer cell type appears to be AML, which is consistent with the previously described CDK8/19 inhibitor corticostatin, suggesting that the compounds which are claimed to be CDK8/19 inhibitors provide the same biological effects. Even in AML, there appears to be little or no therapeutic window, which is relevant since this would likely be the most attractive clinical indication.

I think this is an unusual paper for *eLife*, as there is little new biologically here, per se. However, in the broader context of contributing to our understanding of complex biology of mediator kinases and the potential for therapeutic benefit, I think it does warrant publication in a high profile/general interest journal. As I cited above, many of the papers in this area in the last two years have appeared in similar general interest journals, highlighting the impact this manuscript could have. One always wonders if the authors of such a paper have an "axe" to grind in some way. However, since these authors previously were pursuing this same target class independently, I think they were not initially hoping to invalidate the targets. As the community recently saw with MTH1 "inhibitors" reported by two groups in Nature and then followed up by high quality on target molecules invalidating MTH1, with better tool compounds, it is critical to publish results from testing of second/third generation tool compounds as these clarify the real outcome if such agents would be put in the clinic.

I support publication, but suggest the authors comment more specifically on how readers should judge the likelihood of other transcriptional regulatory kinases such as CDK7, CDK12, and CDK13 having a suitable therapeutic index in cancer. Also, the authors should comment on the corticostatin work as this is very directly comparable.

---

## [Author Response]

*Both reviewers and the editors found your manuscript to be important. The discoveries related to the pharmacological effects of selective Cdk8/19 inhibitors, including their potentially limited therapeutic window, should be of broad interest to the scientific community. The reviewers have recommended some minor revisions to improve the clarity of the manuscript. Please read their comments and revise your manuscript accordingly. As part of this process, we would also like for you to shorten the Impact statement. We do not anticipate that the revised manuscript will require re-review before acceptance.*

We have revised and shortened the impact statement as requested, from:

“Two structurally distinct potent and selective chemical probes for CDK8/19 did not show a clear therapeutic window between exposures required for biomarker inhibition and antitumor activity in experimental tumor models and those causing major pleiotropic toxicity; this complex toxicological profile, associated with on-target inhibition, suggests caution when considering the clinical development of ATP-competitive inhibitors of CDK8/19.’

Now revised to:

“Detailed molecular profiling investigations and tolerability studies in animals suggest caution when considering the clinical development of inhibitors of CDK8 and CDK19, since a clear therapeutic window could not be demonstrated with two structurally distinct, potent and selective compounds”.

*Reviewer #2:*

*[…]I support publication, but suggest the authors comment more specifically on how readers should judge the likelihood of other transcriptional regulatory kinases such as CDK7, CDK12, and CDK13 having a suitable therapeutic index in cancer.*

We are responding to this comment first of all by summarising what we feel are the pertinent published findings and then by summarising the implications and the short paragraph we have inserted into the text. In their Nature paper, Pelish and colleagues (Pelish et al. 2015. Nature 526:273-276) noted that AML cell growth was sensitive to cortistatin A and also a BRD4 inhibitor, I-BET151, that had opposing effects on super-enhancer associated gene expression. They suggested that AML cells are dependent on a ‘precise dosage’ of super-enhancer associated gene expression for survival. Importantly, even though the two compounds have opposing effects, combined treatment showed that I-BET151 had a dominant effect over cortistatin A. This was consistent with super-enhancer genes being highly expressed, but restrained by the mediator complex from even higher expression.

Consistent with the dependence of tumor cells on super-enhancer activity are observations with the recently discovered CDK7-selective inhibitors THZ1 and THZ2. These compounds preferentially decrease expression of super-enhancer associated genes in neuroblastoma and small cell lung cancer (Chipumuro et al. 2014. Cell 159:1126-1139; Christensen et al. 2014. Cancer Cell 26:909-922). A further study in triple-negative breast cancer found a unique dependence on CDK7 for survival that was mediated through an ‘Achilles cluster’ of super-enhancer associated genes (Wang et al. 2015. Cell 163:174-186).

Mouse knockout studies of CDK7 and *MAT1*, a co-factor required for CDK7 activity, have shown that loss of CDK7 activity is embryonically lethal (Rossi et al. 2001. EMBO J. 20:2844-2856; Ganuza et al. 2012 EMBO J. 31:2498-2510). CDK7 was indispensable for embryonic cell proliferation. In adult mice, elimination of CDK7 had phenotypic consequences in those tissues with elevated cell turnover. This resulted in an age-related phenotype with depletion of stem cell pools, telomere shortening and early death.

The recently described THZ531, a covalent inhibitor of CDK12/13, is reported to repress transcription factor and super-enhancer associated gene expression more than expression associated with regular enhancers (Zhang et al. 2016. Nature Chem. Biol. 12:876-884). CDK12 knockout is also embryonically lethal in mice and linked to reduced gene expression of regulators of pluripotency and DNA damage response (Juan et al. 2016. Cell Death Diff. 23:1038–1048).

Overall, inhibitors of the transcriptional kinases CDK7, CDK12 and CDK13 are predicted to have antitumor activity either in the case of CDK7 from exploiting the dependences of certain oncogenic drivers on elevated super-enhancer activity or for CDK12/13 through effects on super-enhancers or regulation of genome instability. Super-enhancers are frequently found associated with genes whose products control the pluripotent state or define cell identity. Therefore targeting transcriptional CDKs such as 7, 12 and 13 may deplete the stem cell compartment, as detected in conditional CDK7 knockouts in adult mice described above. In an *in vivo* model of an oncogenically activated stem cell compartment signalling we report that our CDK8/19 inhibitors altered the proportion of stem cells to proliferative TA cells in the hyperplastic crypt that may in part be due to super-enhancer activation. This raises the hypothetical possibility that similar to AML cells, the stem cell compartment requires a precise ‘dosage’ of super-enhancer activity and that activation or inhibition of super-enhancer activity will have a negative impact on the stem cell compartment. The selective CDK7 tool inhibitors are reported to be tolerated in mice and so are predicted to be non-toxic; however, as we reported, our CDK8/19 inhibitors were also tolerated in mice and the toxicity was not revealed until detailed tolerability studies were performed in higher species. This suggests that the true toxicity profile of inhibitors of the other transcriptional CDKs may not be revealed until dedicated tolerability studies are performed in higher species using selective compounds with optimised pharmaceutical properties. We have added a short paragraph to the Discussion to cover these points (eighth paragraph).

*Also, the authors should comment on the corticostatin work as this is very directly comparable.*

Pelish and colleagues (Pelish et al. 2015. Nature 526:273-276) found AML cells were particularly sensitive to cortistatin A treatment and demonstrated corresponding CDK8/19 inhibition with STAT1^SER727^, a target engagement biomarker for CDK8/19 inhibition. They also demonstrated that Cortistatin A inhibited the CDK8 kinase module, but did not inhibit other transcriptional CDKs and did not bind to ROCK1 or 2, the other described targets for cortistatin A, in MOLM-4 AML cell lysates. In addition they showed that Cortistatin A was orally bioavailable, well tolerated in mice and efficacious against AML solid tumor xenograft or systemic disease models. The authors concluded that ‘The specificity, potency, favorable pharmacokinetics and long residence time of CA make it a useful *in vitro* and *in vivo* probe of Mediator kinases and a promising lead for development of therapeutics.’ In our paper we employed two different and chemically distinct series of potent and selective CDK8/19 modulators s that are much less challenging to synthesise compared to cortistatin A. These compounds had optimal pharmacological and pharmaceutical properties with single digit nM affinities for CDK8/19 and low nanomolar activity in promoter and STAT1^SER727^ reporter assays and are as potent as cortistatin A. As described by Pelish and colleagues for cortistatin A, we found similar induction of super-enhancer-regulated genes, although we carried out gene expression profiling of xenograft tumors treated *in vivo* rather than *in vitro* culture. Our compounds were similarly tolerated in mice and we also independently found AML to be a highly sensitive tumor type for our compounds. However, in our study we also tested our two series in dedicated tolerability studies in higher species that revealed toxicity not apparent in mouse studies. Given the similarities in the *in vitro* and *in vivo* studies between cortistatin A and our own two series, we predict that cortistatin A will exhibit similar toxicity.

We already refer to cortistatin A at various points in the Discussion; however, in response to reviewer 2 we have modified the text to further address the comparison between our compounds and that described for cortistatin A and to indicate that Cortistatin A is likely to share similar tolerability problems (Discussion, sixth paragraph).